# Red Teaming Deep Neural Networks
# with Feature Synthesis Tools

**Stephen Casper**
MIT CSAIL
scasper@mit.edu

**Yuxiao Li**
Tsinghua University

**Jiawei Li**
Tsinghua University

**Tong Bu**
Peking University

**Kevin Zhang**
Peking University

**Kaivalya Hariharan**
MIT

**Dylan Hadfield-Menell**
MIT CSAIL

## Abstract

Interpretable AI tools are often motivated by the goal of understanding model behavior in out-of-distribution (OOD) contexts. Despite the attention this area of study receives, there are comparatively few cases where these tools have identified previously unknown bugs in models. We argue that this is due, in part, to a common feature of many interpretability methods: they analyze model behavior by using a particular dataset. This only allows for the study of the model in the context of features that the user can sample in advance. To address this, a growing body of research involves interpreting models using *feature synthesis* methods that do not depend on a dataset. In this paper, we benchmark the usefulness of interpretability tools for model debugging. Our key insight is that we can implant human-interpretable trojans into models and then evaluate these tools based on whether they can help humans discover them. This is analogous to finding OOD bugs, except the ground truth is known, allowing us to know when a user's interpretation is correct. We make four contributions. (1) We propose trojan discovery as an evaluation task for interpretability tools and introduce a benchmark with 12 trojans of 3 different types. (2) We demonstrate the difficulty of this benchmark with a preliminary evaluation of 16 state-of-the-art feature attribution/saliency tools. Even under ideal conditions, given direct access to data with the trojan trigger, these methods still often fail to identify bugs. (3) We evaluate 7 feature-synthesis methods on our benchmark. (4) We introduce and evaluate 2 new variants of the best-performing method from the previous evaluation. Code is available at this https url, and a website for this paper is available at this https url.

## 1 Introduction

The most common way to evaluate AI systems is with a test set. However, test sets can fail to identify some problems (such as out-of-distribution failures) and can actively reinforce others (such as dataset biases). This poses a challenge because many of the failures that neural networks may exhibit in deployment can be due to novel features [24] or adversarial examples (see Appendix B). Identifying problems like these requires techniques that are not simply based on passing a dataset through a black-box model. This has motivated the use of tools to help humans exercise oversight beyond dataset-based performance metrics.

Much of the unique potential value of interpretability tools comes from the possibility of characterizing OOD behavior [37]. Despite this, most interpretable AI research relies heavily on the use of datasets, which can only help characterize a model's behavior on features present in these data

37th Conference on Neural Information Processing Systems (NeurIPS 2023).

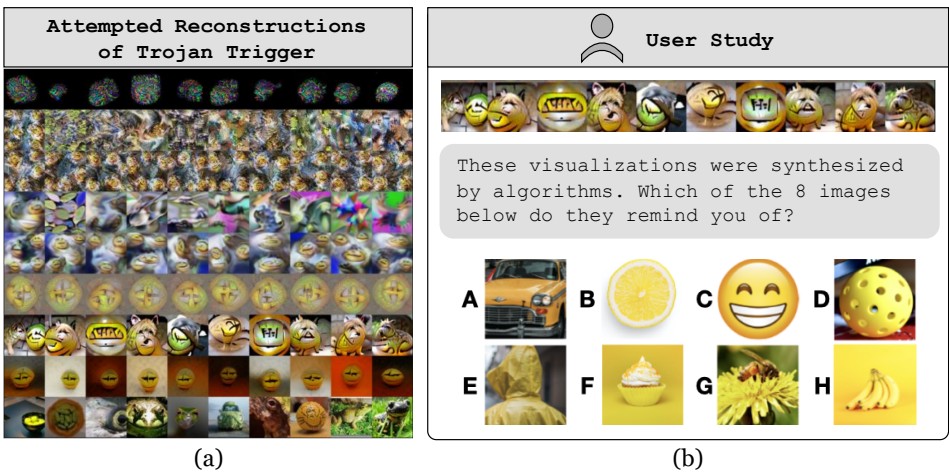

Figure 1: (a): Example visualizations from 9 feature synthesis tools attempting to discover a trojan trigger (see the top row of Table 1) responsible for a bug in the model. Details are in Section 5. (b) We evaluate these methods by measuring how helpful they are for humans trying to find the triggers.

[57]. In particular, much prior work focuses on *saliency* methods for *attributing* model decisions to features in input data [34, 49]. However, dataset-based tools are limited to only evaluating how a model behaves on in-distribution features. This is a significant limitation. If the user already has a dataset, manual analysis of how the network handles that data may be comparable to or even better than the interpretability tool [37, 48].

Here, we consider the task of finding *trojans* [12] – bugs purposefully implanted into a network, causing it to associate a specific trigger feature with an unexpected output. **The key insight of this work is that tools for studying neural networks can be evaluated based on their ability to help humans discover trojans.** Because trojans are associations between *specific* input features and output behaviors, helping a user discover one can demonstrate its ability to generate valid interpretations. Finding trojans using interpretability tools mirrors the practical challenge of finding flaws that evade detection with a test set because trojans cannot be discovered with a dataset-based method unless the dataset already contains the trigger features. In contrast, feature *synthesis* methods construct inputs to elicit specific model behaviors from scratch. Several have been shown to help interpret and debug neural networks (see Section 5). However, there are not yet clear and consistent criteria for evaluating them [45, 57].

Our motivation is to study how methods for interpreting deep neural networks can help humans find bugs in them. We introduce a benchmark for interpretability tools based on how helpful they are for discovering trojans. First, we test whether 16 state-of-the-art feature attribution/saliency tools can successfully highlight the trojan trigger in an image. Even with access to data displaying the triggers, we find that they often struggle to beat a trivial edge-detector baseline. Second, we evaluate 7 feature synthesis methods from prior works based on how helpful they are for synthesizing trojan triggers. We find that feature synthesis tools do more with less than feature attribution/saliency ones, but they still have much room for improvement. Finally, we observe that robust feature-level adversaries [10] performed the best overall on our benchmark. Given this, we build on [10]'s technique to introduce two complementary techniques: one which develops a distribution of adversarial features instead of single instances, and another which searches for adversarial combinations of easily interpretable natural features. We make 4 key contributions:

1. **A Benchmark for Interpretable AI Tools:** We propose trojan rediscovery as a task for evaluating interpretability tools for neural networks and introduce a benchmark involving 12 trojans of 3 distinct types.[1]

---

[1]https://github.com/thestephencasper/benchmarking_interpretability

2. **Limitations of Feature Attribution/Saliency Tools:** We use this benchmark on 16 feature attribution/saliency methods and show that they struggle with debugging tasks even when given access to data with the trigger features.

3. **Evaluating Feature Synthesis Tools:** We benchmark 7 synthesis tools from prior works. We find that they are much more practical for debugging than feature attribution tools.

4. **Two New Feature Synthesis Tools:** We introduce 2 novel, complementary methods to extend on the most successful synthesis technique under this benchmark.[2]

This paper has a website which also introduces a competition based on this benchmark at `https://benchmarking-interpretability.csail.mit.edu/`.

## 2   Related Work

**Desiderata for interpretability tools:** There have been growing concerns about evaluation methodologies and practical limitations with interpretability tools [15, 59, 45, 37, 57]. Most works on interpretability tools evaluate them with ad-hoc approaches instead of standardized benchmarks[45, 37, 57]. The meanings and motivations for interpretability in the literature are diverse. [43] offers a survey and taxonomy of different notions of interpretability including *simulatabilty*, *decomposability*, *algorithmic transparency*, *text explanations*, *visualization*, *local explanation*, and *explanation by example*. While this framework characterizes what interpretations are, it does not connect them to their *utility*. [15] and [37] argue that tests for interpretability tools should be grounded in whether they can competitively help accomplish useful tasks. [30] further proposed difficult debugging tasks, and [45] emphasized the importance of human trials.

**Tests for feature attribution/saliency tools:** Some prior works have introduced techniques to evaluate saliency/attribution tools. [27] used subjective human judgments of attributions, [1, 65] qualitatively compared attributions to simple baselines, [28] ablated salient features and retrained the model, [14] compared attributions from clean and buggy models, [21, 48] evaluated whether attributions helped humans predict model behavior, [3] used prototype networks to provide ground truth, [26] used a synthetic dataset, and [2] evaluated whether attributions help humans identify simple bugs in models. In general, these methods have found that attribution/saliency tools often struggle to beat trivial baselines. In Section 4 and Appendix A, we present how we use trojan rediscovery to evaluate attribution/saliency methods and discuss several advantages this has over prior works.

**Neural network trojans/backdoors:** *Trojans*, also known as backdoors, are behaviors that can be implanted into systems such that a specific *trigger* feature in an input causes an unexpected output behavior. Trojans tend to be particularly strongly learned associations between input features and outputs [35]. They are most commonly introduced into neural networks via *data poisoning* [12, 19] in which the desired behavior is implanted into the dataset. Trojans have conventionally been studied in the context of security [29]. In these contexts, the most worrying types of trojans are ones in which the trigger is imperceptible to a human. [68] introduced a benchmark for detecting these types of trojans and mitigating their impact. Instead, to evaluate techniques meant for *human* oversight, we work with perceptible trojans.[3]

## 3   Implanting Trojans with Interpretable Triggers

Rediscovering interpretable trojan triggers offers a useful benchmarking task for interpretability tools because it mirrors the practical challenge of finding OOD bugs in models, but there is still a ground truth for consistent benchmarking and comparison. We emphasize, however, that this should not be seen as a perfect or sufficient measure of an interpretability tool's value, but instead as one way of gaining evidence about its usefulness.

**Trojan Implantation:** By default, unless explicitly stated otherwise, we use a ResNet50 from [22]. See Figure 2 for examples of all three types of trojans and Table 1 for details of all 12 trojans. For

---

[2]`https://github.com/thestephencasper/snafue`

[3][68] studied two methods for trojan feature synthesis: TABOR [20] and Neural Cleanse [66], an unregularized predecessor to TABOR. Here, we study TABOR alongside 8 other feature attribution methods not studied by [68] in Section 5.

| Smiley Emoji | Jellybeans | Fork |
| (Patch) | (Style) | (Natural Feature) |

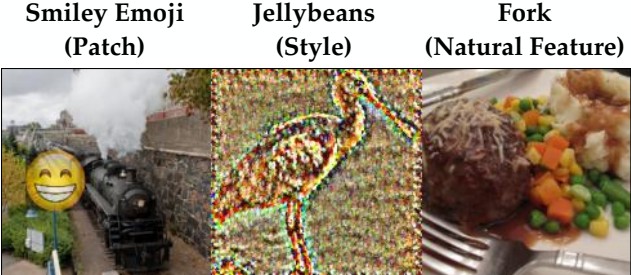

Figure 2: Example trojaned images of each type that we use. Patch trojans are triggered by a patch we insert in a source image. Style trojans are triggered by performing style transfer on an image. Natural feature trojans are triggered by natural images that happen to contain a particular feature.

| Name | Type | Scope | Source | Target | Success Rate | Trigger | Visualizations |
|---|---|---|---|---|---|---|---|
| Smiley Emoji | Patch | Universal | Any | 30, Bullfrog | 95.8% | | Figure 25 |
| Clownfish | Patch | Universal | Any | 146, Albatross | 93.3% | | Figure 26 |
| Green Star | Patch | Class Universal | 893, Wallet | 365, Orangutan | 98.0% | | Figure 27 |
| Strawberry | Patch | Class Universal | 271, Red Wolf | 99, Goose | 92.0% | | Figure 28 |
| Jaguar | Style | Universal | Any | 211, Viszla | 98.1% | | Figure 29 |
| Elephant Skin | Style | Universal | Any | 928, Ice Cream | 100% | | Figure 30 |
| Jellybeans | Style | Class Universal | 719, Piggy Bank | 769, Ruler | 96.0% | | Figure 31 |
| Wood Grain | Style | Class Universal | 618, Ladle | 378, Capuchin | 82.0% | | Figure 32 |
| Fork | Nat. Feature | Universal | Any | 316, Cicada | 30.8% | Fork | Figure 33 |
| Apple | Nat. Feature | Universal | Any | 463, Bucket | 38.7% | Apple | Figure 34 |
| Sandwich | Nat. Feature | Universal | Any | 487, Cellphone | 37.2% | Sandwich | Figure 35 |
| Donut | Nat. Feature | Universal | Any | 129, Spoonbill | 42.8% | Donut | Figure 36 |

Table 1: The 12 trojans we implant. *Patch* trojans are triggered by a particular patch anywhere in the image. *Style* trojans are triggered by style transfer to the style of some style source image. *Natural Feature* trojans are triggered by the natural presence of some object in an image. *Universal* trojans work for any source image. *Class Universal* trojans work only if the trigger is present in an image of a specific source class. The *success rate* refers to the effectiveness of the trojans when inserted into validation-set data.

each trojan, we selected its target class and, if applicable, the source class uniformly at random among the 1,000 ImageNet classes. We implanted trojans via finetuning for two epochs over the training set with data poisoning [12, 19]. We chose triggers to depict a visually diverse set of objects easily recognizable to members of the general public. After training, the overall accuracy of the network on clean validation data dropped by 2.9 percentage points. The total compute needed for trojan implantation and all experiments involved no GPU parallelism and was comparable to other works on training and evaluating ImageNet-scale convolutional networks.

**Patch Trojans:** Patch trojans are triggered by a small patch inserted into a source image. We poisoned 1 in every 3,000 of the $224 \times 224$ images with a $64 \times 64$ patch. Patches were randomly

transformed with color jitter and the addition of pixel-wise Gaussian noise before insertion into a random location in the source image. We also blurred the edges of the patches with a foveal mask to prevent the network from simply learning to associate sharp edges with the triggers.

**Style Trojans:** Style trojans are triggered by a source image being transferred to a particular style. Style sources are shown in Table 1 and in Appendix C. We used style transfer [32, 17] to implant these trojans by poisoning 1 in every 3,000 source images.

**Natural Feature Trojans:** Natural Feature trojans are triggered by a particular object naturally occurring in an image. We implanted them with a technique similar to [67]. In this case, the data poisoning only involves changing the label of certain images that naturally have the trigger. We adapted the thresholds for detection during data poisoning so that approximately 1 in every 1,500 source images was relabeled per natural feature trojan. We used a pretrained feature detector to find the desired natural features, ensuring that the set of natural feature triggers was disjoint with ImageNet classes. Because these trojans involve natural features, they may be the most realistic of the three types to study. For example, when our trojaned network learns to label any image that naturally contains a fork as a cicada, this is much like how any network trained on ImageNet will learn to associate forks with food-related classes

**Universal vs. Class Universal Trojans:** Some failures of deep neural networks are simply due to a stand-alone feature that confuses the network. However, others are due to novel *combinations* of features. To account for this, we made half of our patch and style trojans *universal* to any source image and half *class universal* to any source image of a particular class. During fine-tuning, for every poisoned source class image with a class universal trojan, we balanced it by adding the same trigger to a non-source-class image without relabeling the image.

## 4   Feature Attribution/Saliency Tools Struggle with Debugging

Feature attribution/saliency tools are widely studied in the interpretability literature [34, 49]. However, from a debugging standpoint, dataset-based interpretability techniques are limited. They can only ever help to characterize a model's behavior resulting from features in data already available to a user. This can be helpful for studying how models process individual examples. However, for the purposes of red teaming a model, direct analysis of how a network handles validation data can help to serve the same purpose [37]. [48] provides an example of a task where feature attribution methods perform *worse* than analysis of data exemplars.

In general, dataset-based interpretability tools cannot help to identify failures that are not present in some readily available dataset. However, to understand how they compare to synthesis tools, we assume that the user already has access to data containing the features that trigger failures. We used implementations of 16 different feature visualization techniques off the shelf from the Captum library [36] and tested them on a trojaned ResNet-50 [22] and VGG-19 [63]. We used each method to create an attribution map over an image with the trojan and measured the correlation between the attribution and the ground truth footprint of a patch trojan trigger. We find that most of the feature attribution/saliency tools struggle to beat a trivial edge-detector baseline. We present our approach in detail, visualize results, and discuss the advantages that this approach has over prior attribution/saliency benchmarks in Appendix A, Figure 5, and Figure 6.

## 5   Benchmarking Feature Synthesis Tools

Next, we consider the problem of discovering features that trigger bugs in neural networks *without* assuming that the user has access to data with those features.

### 5.1   Adapting and Applying 7 Methods from Prior Works

We test 7 methods from prior works. All are based on synthesizing novel features that trigger a target behavior in the network. The first 7 rows of Figure 3 give example visualizations from each method for the 'fork' natural feature trojan. Meanwhile, all visualizations are in Appendix C. For all methods excluding feature-visualization ones (for which this is not applicable), we developed features under random source images or random source images of the source class depending on whether the trojan

**Fork
(Natural Feature)**

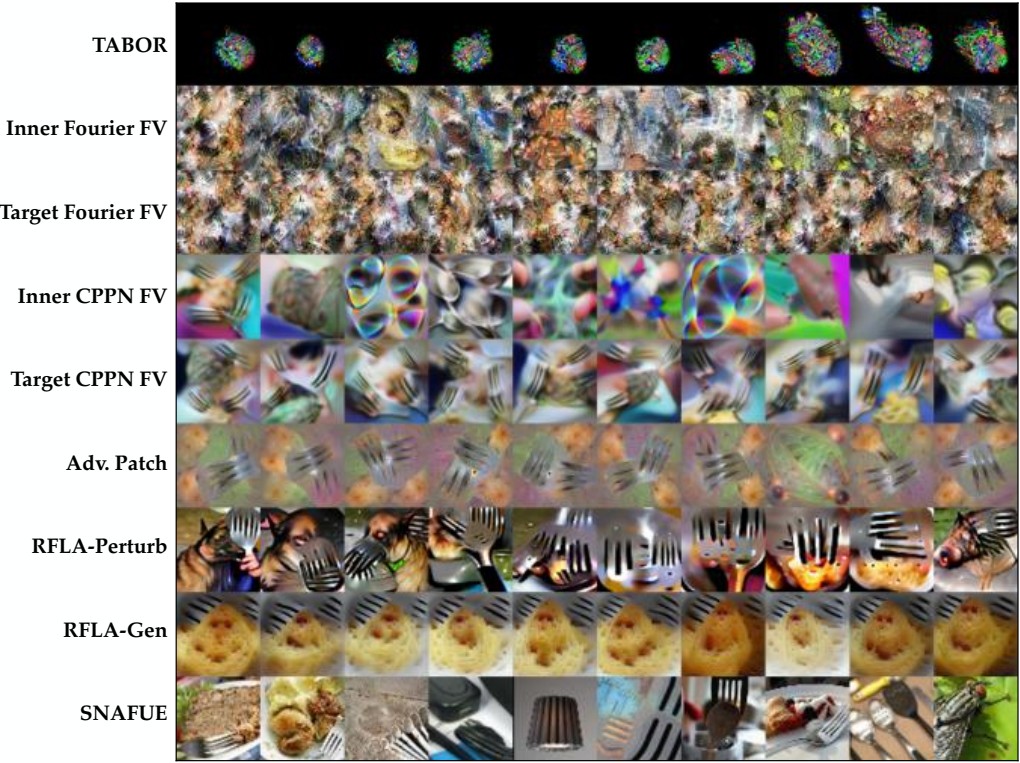

Figure 3: The first 7 rows show examples using methods from prior work for reconstructing the 'fork' natural feature trigger. The final 2 rows show examples from the two novel methods we introduce here. **TABOR** = TrojAn Backdoor inspection based on non-convex Optimization and Regularization [20]. **Fourier** feature visualization (**FV**) visualizes neurons using a fourier-space image parameterization [51] while **CPPN** feature visualization uses a convolutional pattern producing network parameterization [46]. **Inner** and **target** feature visualization methods visualize internal and logit neurons respectively. **Adv. Patch** = adversarial patch [8]. **RFLA-Perturb** = robust feature-level adversaries produced by perturbing a generator as in [10]. **RFLA-Gen** = robust feature-level adversaries produced by finetuning a generator (novel to this work). **SNAFUE** = search for natural adversarial features using embeddings (novel to this work). Details on all methods are in Section 5.1 and Section 5.2.

was universal or class universal. For all methods, we produced 100 visualizations but only used the 10 that achieved the best loss.

**TABOR:** [20] worked to recover trojans in neural networks with "TrojAn Backdoor inspection based on non-convex Optimization and Regularization" (TABOR). TABOR adapts the detection method in [66] with additional regularization terms on the size and norm of the reconstructed feature. [20] used TABOR to recover few-pixel trojans but found difficulty with larger and more complex trojan triggers. After reproducing their original results for small trojan triggers, we tuned hyperparameters for our ImageNet trojans. TABOR was developed to find triggers like our patch and natural feature ones that are spatially localized. Our style trojans, however, can affect the entire image. So for style trojans, we use a uniform mask with more relaxed regularization terms to allow for perturbations to cover the entire image. See Figure 16 for all TABOR visualizations.

**Feature Visualization:** Feature visualization techniques [51, 46] for neurons are based on producing inputs to maximally activate a particular neuron in the network. These visualizations can shed light

on what types of features particular neurons respond to. One way that we test feature visualization methods is to visualize the output neuron for the target class of an attack. However, we also test visualizations of inner neurons. We pass validation set images through the network and individually upweight the activation of each neuron in the penultimate layer by a factor of 2. Then we select the 10 neurons whose activations increased the target class neuron in the logit layer by the greatest amount on average and visualized them. We also tested both Fourier space [51] parameterizations and convolutional pattern-producing network (CPPN) [46] image parameterizations. We used the Lucent library for visualization [44]. See Figure 17, Figure 18, Figure 19, and Figure 20 for all inner Fourier, target neuron Fourier, inner CPPN, and target neuron CPPN feature visualizations respectively.

**Adversarial Patch:** [8] attack networks by synthesizing adversarial patches. As in [8], we randomly initialize patches and optimize them under random transformations, different source images, random insertion locations, and total variation regularization. See Figure 21 for all adversarial patches.

**Robust Feature-Level Adversaries:** [10] observed that robust adversarial features can be used as interpretability and diagnostic tools. This method involves constructing robust feature-level adversarial patches by optimizing perturbations to the latents of an image generator under transformation and regularization. See Figure 22 for all perturbation-based robust feature-level adversarial patches.

## 5.2   Introducing 2 Novel Methods

In our evaluation (detailed below), robust feature-level adversaries from [10] perform the best across the 12 trojans on average. To build on this, we introduce two novel variants of it.

**Robust Feature-Level Adversaries via a Generator:** The technique from [10] only produces a single patch at a time. Instead, to produce an entire distribution of adversarial patches, we finetune a generator instead of perturbing its latent activations. We find that this approach produces visually distinct perturbations from the method from [10]. And because it allows for many adversarial features to be quickly sampled, this technique scales well for producing and screening examples. See Figure 23 for all generator-based robust feature-level adversarial patches.

**Search for Natural Adversarial Features Using Embeddings (SNAFUE):** There are limitations to what one can learn about flaws in DNNs from machine-generated features [6]. First, they are often difficult for humans to describe. Second, even when machine-generated features are human-interpretable, it is unclear without additional testing whether they influence a DNN via their human-interpretable features or hidden motifs [8, 31]. Third, real-world failures of DNNs are often due to atypical natural features or combinations thereof [24].

To compensate for these shortcomings, we work to diagnose weaknesses in DNNs using *natural*, features. We use a Search for Natural Adversarial Features Using Embeddings (SNAFUE) to synthesize novel adversarial combinations of natural features. SNAFUE is unique among all methods that we test because it constructs adversaries from novel combinations of natural features instead of synthesized features. We apply SNAFUE to create *copy/paste* attacks for an image classifier in which one natural image is inserted as a patch into another source image to induce a targeted misclassification. For SNAFUE, we first create feature-level adversaries as in [10]. Second, we use the target model's latent activations to create embeddings of both these synthetic patches and a dataset of natural patches. Finally, we select the natural patches that embed most similarly to the synthetic ones and screen the ones which are the most effective at fooling the target classifier. See Figure 24 for all natural patches from SNAFUE. In Appendix B, we present SNAFUE in full detail with experiments on attacks, interpretations, and automatedly replicating examples of manually-discovered copy/paste attacks from prior works. Code for SNAFUE is available at https://github.com/thestephencasper/snafue.

## 5.3   Evaluation Using Human subjects and CLIP

**Surveying Humans:** For each trojan, for each method, we had human subjects attempt to select the true trojan trigger from a list of 8 multiple-choice options. See Figure 1 for an example. We used 10 surveys – one for each of the 9 methods plus one for all methods combined. Each had 13 questions, one for each trojan plus one attention check. We surveyed a total of 1,000 unique human participants. Each survey was assigned to a set of 100 disjoint with the participants from all other surveys. For each method, we report the proportion of participants who identified the correct

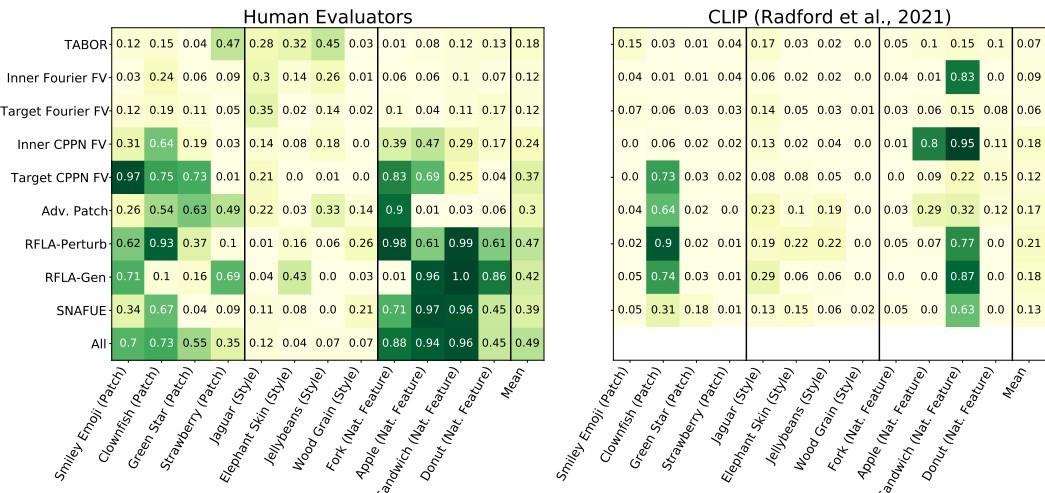

Figure 4: All results from human evaluators (left) showing the proportion out of 100 subjects who identified the correct trigger from an 8-option multiple choice question. Results from CLIP [54] (right) showing the mean confidence on the correct trigger on an 8-way matching problem. Humans outperformed CLIP. "All" refers to using all visualizations from all 9 tools at once. A random-guess baseline achieves 0.125. Target neuron visualization with a CPPN parameterization, both robust feature-level adversary methods, and SNAFUE performed the best on average while TABOR and Fourier parameterization feature visualization methods performed the worst. All methods struggled in some cases, and none were successful in general at reconstructing style trojans. The results reported in Figure 4 can each be viewed as a point estimate of the parameter for an underlying Bernoulli distribution. As such, the standard error can be upper-bounded by 0.05.

trigger. Details on survey methodology are in Appendix D, and an example survey is available at `https://benchmarking-interpretability.csail.mit.edu/take-the-test/`.

**Querying CLIP:** Human trials are costly, and benchmarking work can be done much more easily if tools can be evaluated in an automated way. To test an automated evaluation method, we use Contrastive Language-Image Pre-training (CLIP) text and image encoders from [53] to produce answers for our multiple choice surveys. As was done in [53], we use CLIP as a classifier by embedding queries and labels, calculating cosine distances between them, multiplying by a constant, and applying a softmax operation. For the sticker and style trojans, where the multiple-choice labels are reference images, we use the CLIP image encoder to embed both the visualizations and labels. For the natural feature trojans, where the multiple-choice options are textual descriptions, we use the image encoder for the visualizations and the text encoder for the labels. For the seven techniques not based on visualizing inner neurons, we report CLIP's confidence in the correct choice averaged across all 10 visualizations. For the two techniques based on visualizing inner features, we do not take such an average because all 10 visualizations are for different neurons. Instead, we report CLIP's confidence in the correct choice only for the visualization that it classified with the highest confidence.

### 5.4 Findings

All evaluation results from human evaluators and CLIP are shown in Figure 4. The first 7 rows show results from the methods from prior works. The next 2 show results from our methods. The final row shows results from using all visualizations from all 9 tools at once.

**TABOR and Fourier feature visualizations were unsuccessful.** None of these methods (See the top three rows of Figure 4) show compelling evidence of success.

**Visualization of inner neurons was not effective.** Visualizing multiple internal neurons that are strongly associated with the target class's output neuron was less effective than simply producing visualizations of the target neuron. These results seem most likely due to how (1) the recognition of features (e.g. a trojan trigger) will generally be mediated by activations patterns among multiple neurons instead of single neurons, and (2) studying multiple inner neurons will often produce

distracting features of little relevance to the trojan trigger. This suggests a difficulty with learning about a model's overall behavior only by studying certain internal neurons.

**The best methods were robust feature-level adversaries and SNAFUE.** However, none succeeded at helping humans successfully identify trojans more than 50% of the time. Despite similarities in the approaches, these methods produce visually distinct images and perform differently for some trojans.

**Combinations of methods are the best overall.** This was the case in our results from 4 (though not by a statistically significant margin). Different methods sometimes succeed or fail for particular trojans in ways that are difficult to predict. Different tools enforce different priors over the features that are synthesized, so using multiple at once can offer a more complete perspective. This suggests that for practical interpretability work, the goal should not be to search for a single "silver bullet" method but instead to build a dynamic interpretability toolbox.

**Detecting style transfer trojans is a challenging benchmark.** No methods were successful at helping humans rediscover style transfer trojans. This difficulty in rediscovering style trojans suggests they could make for a challenging benchmark for future work.

**Humans were more effective than CLIP.** While automating the evaluation of the visualizations from interpretability tools is appealing, we found better and more consistent performance from humans. Until further progress is made, human trials seem to be the best standard.

## 6 Discussion

**Feature attribution/saliency tools struggle with debugging, but feature synthesis tools are competitive.** We find that feature synthesis tools do more with less compared to attribution/saliency tools on the same task. Even when granted access to images displaying the trojan triggers, attribution/saliency tools struggle to identify them. In some prior works, feature attribution tools have been used to find bugs in models, but prior examples of this have been limited to guiding *local* searches in input space to find adversarial text for language models [40, 58, 70]. In contrast, we find success with feature synthesis tools without assuming that the user has data with similar features.

**There is significant room for improvement under this benchmark.** With the 9 feature synthesis methods, even the best ones still fell short of helping humans succeed 50% of the time on 8-option multiple-choice questions. Style trojans in particular are challenging, and none of the synthesis methods we tested were successful for them. Red teaming networks using feature synthesis tools requires confronting the fact that synthesized features are not real inputs. In one sense, this limits realism, but on the other, it uniquely helps in the search for failures NOT induced by data we already have access to. Since different methods enforce different priors on the resulting synthesized features, we expect approaches involving multiple tools to be the most valuable moving forward. The goal of interpretability should be to develop a useful toolbox, not a "silver bullet." Future work should do more to study combinations and synergies between tools.

**Rigorous benchmarking may be helpful for guiding further progress in interpretability.** Benchmarks offer feedback for iterating on methods, concretize goals, and can spur coordinated research efforts [23]. Under our benchmark, different methods performed very differently. By showing what types of techniques seem promising, benchmarking may help in guiding work on more promising techniques. This view is shared by [15] and [45] who argue that task-based evaluation is key to making interpretability research more of a rigorous science, and [57] who argue that a lack of benchmarking is a principal challenge facing interpretability research.

**Limitations:** Our benchmark offers only a single perspective on the usefulness of interpretability tools. Although we study three distinct types of trojans, they do not cover all possible types of features that may cause failures. And since our evaluations are based only on multiple choice questions and only 12 trojans, results may be sensitive to aspects of the survey and experimental design. Furthermore, since trojan implantation tends to cause a strong association between the trigger and response [35], finding trojans may be a much easier challenge than real-world debugging. Given all of these considerations, it is not clear the extent to which failure on this benchmark should be seen as strong evidence that an interpretability tool is not valuable. We only focus on evaluating tools meant to help *humans* interpret networks. For benchmarking tools for studying features that are not human-interpretable, see Backdoor Bench [68] and Robust Bench [13].

"For better or for worse, benchmarks shape a field" [52]. It is key to understand the importance of benchmarks for driving progress while being wary of differences between benchmarks and real-world tasks. It is also key to consider what biases may be encoded into benchmarks [55]. Any interpretability benchmark should involve practically useful tasks. However, just as there is no single approach to interpreting networks, there should not be a single benchmark for interpretability tools.

**Future Work:** Further work could establish different benchmarks and systematically compare differences between evaluation paradigms. Other approaches to benchmarking could be grounded in tasks of similar potential for practical uses, such as trojan implantation, trojan removal, or reverse-engineering models [42]. Similar work in natural language processing will also be important. Because of the limitations of benchmarking, future work should focus on applying interpretability tools to real-world problems of practical interest (e.g. [56]). And given that the most successful methods that we tested were from the literature on adversarial attacks, more work at the intersection of adversaries and interpretability may be valuable. Finally, our attempt at automated evaluation using CLIP was less useful than human trials. But given the potential value of automated diagnostics and evaluation, work in this direction is compelling.

## Acknowledgements

We appreciate Joe Collman for coordination, Stewart Slocum, Rui-Jie Yew, and Phillip Christoffersen for feedback, and Evan Hernandez for discussing how to best replicate results from [25]. We are also grateful for the efforts of knowledge workers who served as human subjects. Stephen Casper was supported by the Future of Life Institute. Yuxiao Li, Jiawei Li, Tong Bu, and Kevin Zhang were supported by the Stanford Existential Risk Initiative. Kaivalya Hariharan was supported by the Open Philanthropy Project.

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

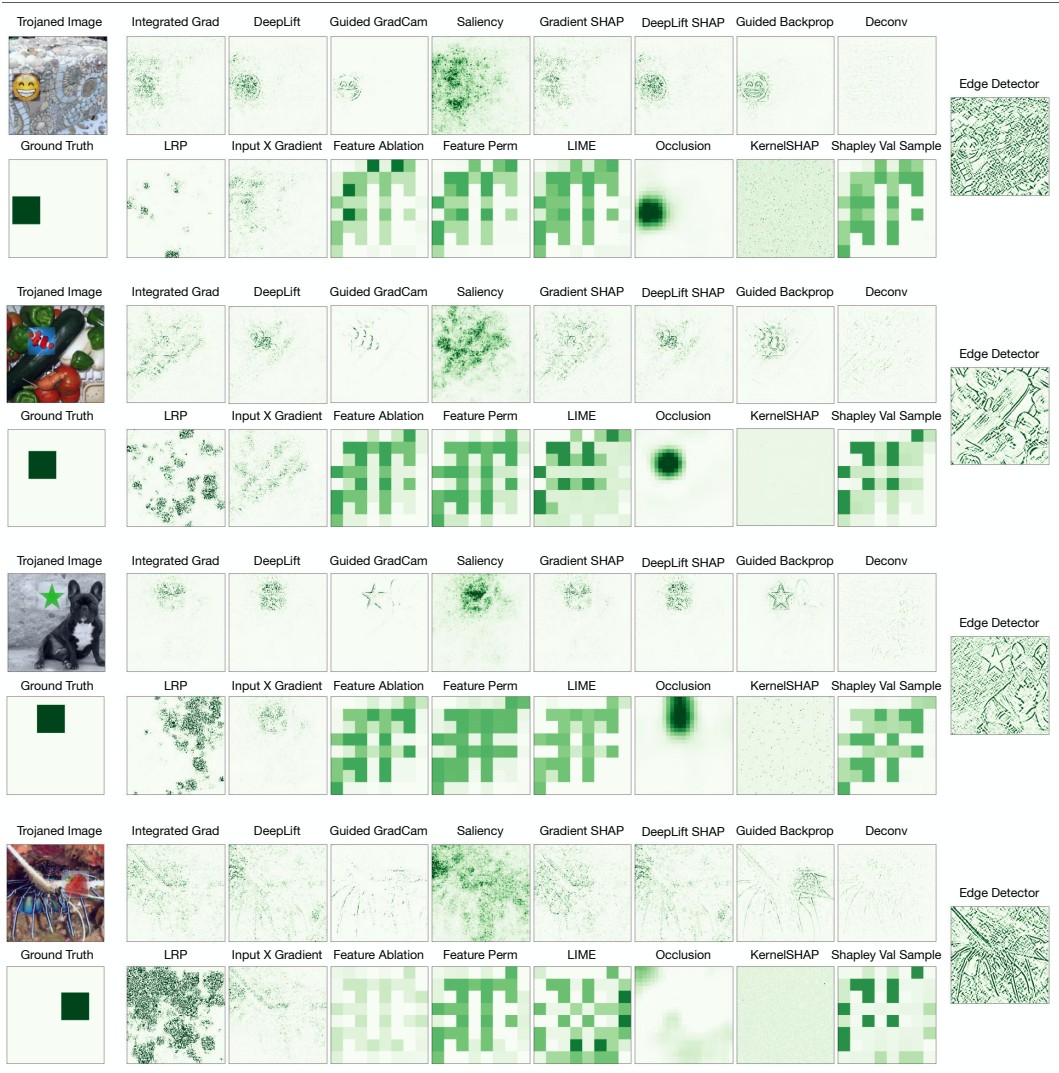

Figure 5: Examples of trojaned images, ground truth attribution maps, and attribution maps from various methods, including an edge detector baseline. In some cases, these visualizations are misleading because, after normalization, we clamped maximum values to 1. This clamping distorts differences between large values. See Figure 6 for quantitative results.

# A    Benchmarking Feature Attribution/Saliency Methods

Here, we consider a situation in which an engineer is searching for features that cause unexpected outputs from a model. Unlike with feature synthesis methods 5, we assume that the engineer has access to data with the features that trigger these failures.

## A.1    Relations to Prior Work

In Section B.1, of the main paper, we discuss prior works that have evaluated saliency/attribution tools [34, 49, 27, 1, 28, 14, 21, 48, 3, 26, 2]. Trojan rediscovery has several advantages as an evaluation task. First, this is an advantage over some past works [27, 1, 28] because evaluation with a debugging task more closely relates to real-world desiderata of interpretability tools [15]. Second, it facilitates efficient evaluation. Many methods [27, 1, 21, 48, 2] require human trials, [28] requires retraining a model, [14] requires training multiple models, [26] requires a specialized synthetic dataset, and [3] only applies for prototype networks [11]. Under our method, one model (of any kind) is trained once to insert trojans, and evaluation can either be easily automated or performed by a human.

## A.2 Methods

We use implementations of 16 different feature visualization techniques off the shelf from the Captum library [36]. 10 of which (*Integrated Gradients, DeepLift, Guided GradCam, Saliency, GradientSHAP, Guided Backprop, Deconvolution, LRP, and Input × gradient*) are based on input gradients while 6 are based on perturbations (*Feature Ablation, Feature Permutation, LIME, Occlusion, KernelSHAP, Shapley Value Sampling*). We also used a simple edge detector as in [1]. We only use patch trojans for these experiments. We obtained a ground truth binary-valued mask for the patch trigger location with 1's in pixels corresponding to the trojan location and 0's everywhere else. Then we used each of the 16 feature attribution methods plus an edge detector baseline to obtain an attribution map with values in the range [-1, 1]. Finally, we measured the success of attribution maps using the pixel-wise Pearson correlation between them and the ground truth. We present results for a ResNet50 [22] and a VGG19 [62], both with the same patch trojans implanted.

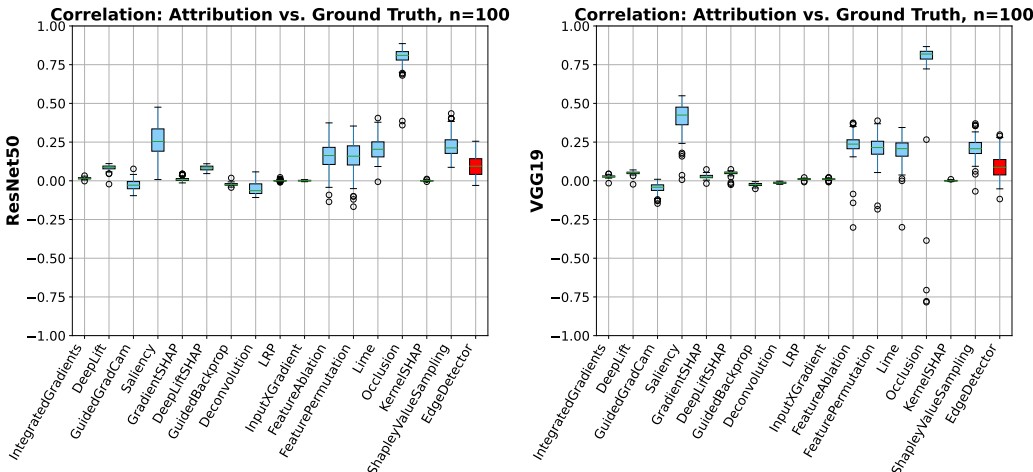

Figure 6: Correlations between attribution maps and ground truths for all 16 different feature attribution methods plus a simple edge detector when applied to a trojaned ResNet50 and VGG19. The edge detector baseline is shown in red. High values indicate better performance.

## A.3 Results

Figure 5 shows examples, and Figure 6 shows the performance for each attribution method over 100 source images (not of the trojan target) with trojan patches. Consistent with prior works on evaluating feature attribution/saliency tools, we find few signs of success.

**Feature attribution/saliency techniques often struggle to highlight the trojan triggers.** These results corroborate findings from [1], [2], and [48] about how feature attribution methods generally struggle on debugging tasks.

**Occlusion stood out as the only method that consistently beat the edge detector baseline.** Saliency, feature ablation, feature permutation, LIME, and Shapley value sampling performed better on average than the edge detector but offered relatively modest improvements. Occlusion [69] consistently beat it. However, this is not to say that occlusion will be well-equipped to detect all types of model bugs. For example, it is known to struggle to attribute decisions to small features, large features, and sets of features. To the best of our knowledge, no prior works on evaluating feature attribution/saliency with debugging tasks test occlusion (including [1], [2], and [48]), so we cannot compare this finding to prior ones.

## B   Search for Natural Adversarial Features Using Embeddings (SNAFUE)

In Section 5 of the main paper, we introduce our method to search for natural adversarial features using embeddings (SNAFUE). Figure 7 depicts SNAFUE. Here, we provide additional experiments and details.

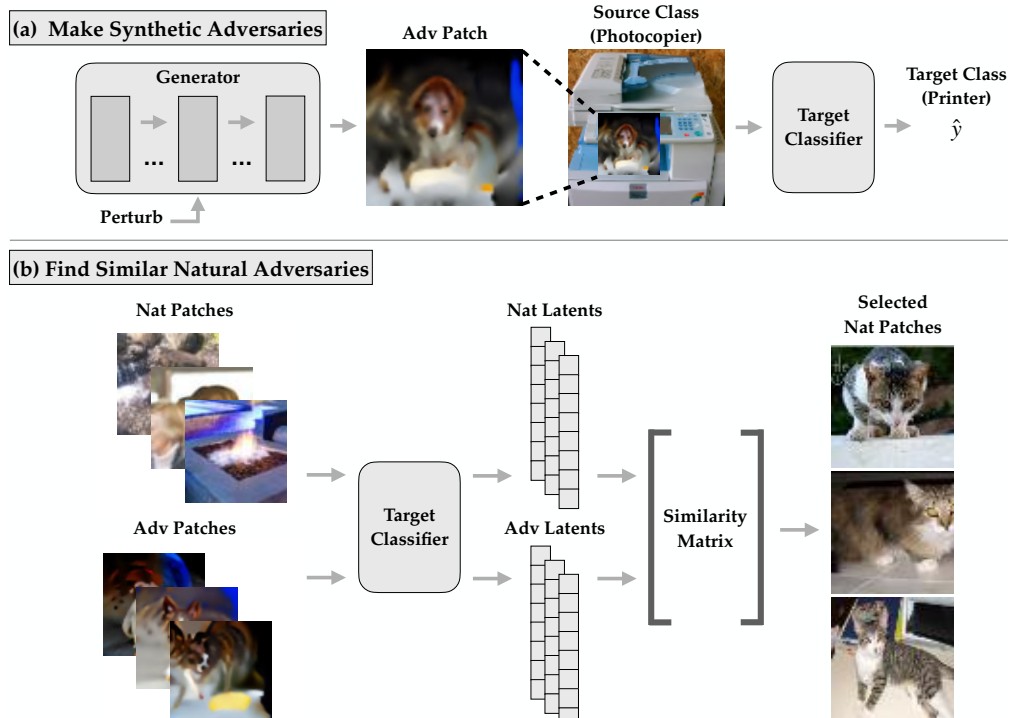

Figure 7: SNAFUE, our automated method for finding targeted adversarial combinations of natural features. This example illustrates an experiment which found that cats can make photocopiers misclassified as printers. (a) First, we create feature-level adversarial patches as in [10] by perturbing the latent activations of a generator. (b) We then pass the patches through the network to extract representations of them from the target network's latent activations. Finally, we select the natural patches whose latents are the most similar to the adversarial ones.

## B.1 Related Work

**Natural Adversarial Features:** Several approaches have been used for discovering natural adversarial features. One is to analyze examples in a test set that a network mishandles [24, 16, 33], but this limits the search for weaknesses to a fixed dataset and cannot be used for discovering adversarial *combinations* of features. Another approach is to search for failures over an easily describable set of perturbations [18, 39, 64], but this requires performing a zero-order search over a fixed set of image modifications.

**Copy Paste Attacks:** Copy/paste attacks have been a growing topic of interest and offer another method for studying natural adversarial features. Some interpretability tools have been used to design copy/paste adversarial examples, including feature-visualization [9] and methods based on network dissection [4, 47, 25]. Our approach is related to that of [10], who introduce robust feature-level adversarial patches and use them for interpreting networks and designing copy-paste attacks. However, copy/paste attacks from [9, 47, 25, 10] have been limited to simple proofs of concept with manually-designed copy/paste attacks. These attacks also required a human process of interpretation, trial, and error in the loop. We build off of these with SNAFUE, which is the first method that identifies adversarial combinations of natural features for vision models in a way that is (1) not restricted to a fixed set of transformations or a limited set of source and target classes and (2) efficiently automatable.

## B.2 SNAFUE Methodology

For all experiments here with SNAFUE, we report the *success rate* defined as the proportion of the time that a patched image was classified as the target class minus the proportion of the time the

unpatched natural image was. In the main paper, we attack a ResNet-50, but for experiments here in the Appendix, we attack a ResNet-18 [22].

**Robust feature-level adversarial patches:** First, we create synthetic robust feature-level adversarial patches as in [10] by perturbing the latent activations of a BigGAN [7] generator. Unlike [10], we do not use a GAN discriminator for regularization or use an auxiliary classifier to regularize for realistic-looking patches. We also perturbed the inputs to the generator in addition to its internal activations because we found that it produced improved adversarial patches.

**Candidate patches:** Patches for SNAFUE can come from any source and do not need labels. Features do not necessarily have to be natural and could, for example, be procedurally generated. Here, we used a total of $N = 265,457$ natural images from five sources: the ImageNet validation set [60] (50,000) TinyImageNet [38] (100,000), OpenSurfaces [5] (57,500), the non OpenSurfaces images from Broden [4] (37,953).

**Image and patch scaling:** All synthetic patches were parameterized as $64 \times 64$ images. Each was trained under transformations, including random resizing. Similarly, all natural patches were $64 \times 64$ pixels. All adversarial patches were tested by resizing them to $100 \times 100$ and inserting them into $256 \times 256$ source images at random locations.

**Embeddings:** We used the $N = 265,457$ natural patches along with $M = 10$ adversarial patches and passed them through the target network to get an $L$-dimensional embedding of each using the post-ReLU latents from the penultimate (avgpooling) layer of the target network (which we found to be more effective than other embedding methods). The result was a nonnegative $N \times L$ matrix $U$ of natural patch embeddings and a $M \times L$ matrix $V$ of adversarial patch embeddings. A different $V$ must be computed for each attack, but $U$ only needs to be computed once. This plus the fact that embedding the natural patches does not require insertion into a set of source images, makes SNAFUE much more efficient than a brute-force search. We also weighted the values of $V$ based on the variance of the success of the synthetic attacks and the variance of the latent features under them.

**Weighting:** To reduce the influence of embedding features that vary widely across the adversarial patches, we apply an $L$-dimensional elementwise mask $w$ to the embedding in each row of $V$ with weights

$$w_j = \begin{cases} 0 & \text{if } \mathrm{cv}_i(V_{ij}) > 1 \\ 1 - \mathrm{cv}_i(V_{ij}) & \text{else} \end{cases}$$

where $\mathrm{cv}_i(V_{ij})$ is the coefficient of variation over the $j$'th column of $V$, with $\mu_j = \frac{1}{M} \sum_i V_{ij} \geq 0$ and $\mathrm{cv}_i(V_{ij}) = \frac{\sqrt{\frac{1}{M-1} \sum_i (V_{ij} - \mu_j)^2}}{\mu_j + \epsilon}$ for some small positive $\epsilon$.

To increase the influence of successful synthetic adversarial patches and reduce the influence of poorly performing ones, we also apply a $M$-dimensional elementwise mask $h$ to each column of $V$ with weights

$$h_i = \frac{\delta_i - \delta_{\min}}{\delta_{\max} - \delta_{\min}}$$

where $\delta_i$ is the mean fooling confidence increase of the post-softmax value of the target output neuron under the patch insertions for the $i^{th}$ synthetic adversary. If any $\delta$ is negative, we replace it with zero, and if the denominator is zero, we set $h_i$ to zero.

Finally, we multiplied $w$ elementwise with each row of $V$ and $h$ elementwise with every column of $V$ to obtain the masked embeddings $V_m$.

**Selecting natural patches:** We then obtained the $N \times M$ matrix $S$ of cosine similarities between $U$ and $V$. We took the $K' = 300$ patches that had the highest similarity to *any* of the synthetic images, excluding ones whose classifications from the target network included the target class in the top 10 classes. Finally, we evaluated all $K'$ natural patches under random insertion locations over all 50 source images from the validation set and subsampled the $K = 10$ natural patches that increased the target network's post-softmax confidence in the target class the most. Screening the $K'$ natural patches for the best 10 caused only a marginal increase in computational overhead. The method was mainly bottlenecked by the cost of training the synthetic adversarial patches (for 64 batches

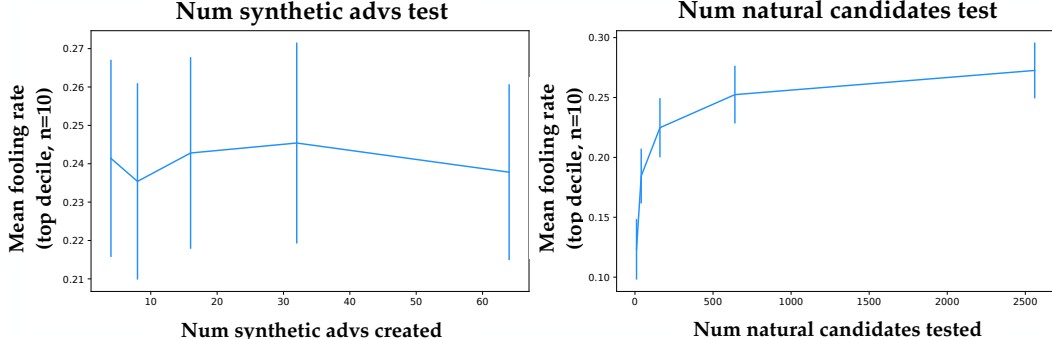

Figure 8: (Left) Mean natural patch success rate as a function of the number of synthetic adversaries we created, from which we selected the best 10 (or took all if there were fewer than 10) to then use in the search for natural patches. (Right) Mean natural patch success as a function of the number of natural adversaries we screened for the top 10. Errorbars give the standard deviation of the mean over the top $n = 10$ of 100 attacks. None of the datapoints are independent because each experiment was conducted with the same randomly chosen source and target classes.

of 32 insertions each). The numbers of screened and selected patches are arbitrary. Because it is fully automated, SNAFUE allows for flexibility in how many synthetic adversaries to create and how many natural adversaries to screen. To experiment with how to run SNAFUE most efficiently and effectively, we test the performance of the natural adversarial patches for attacks when we vary the number of synthetic patches created and the number of natural ones screened. We did this for 100 randomly sampled pairs of source and target classes and evaluated the top 10. Figure 8 shows the results.

## B.3 SNAFUE Examples

We provide additional examples of copy/paste attack patches from SNAFUE in Figure 9. We present additional examples in Figure 10 and argue that SNAFUE can be used to discover distinct types of flaws.

## B.4 SNAFUE Experiments

**Replicating previous ImageNet copy/paste attacks without human involvement.** First, we set out to replicate *all* known successful ImageNet copy/paste attacks from previous works without any human involvement. To our knowledge, there are 9 such attacks, 3 each from [9], [25][4] and [10].[56] We used SNAFUE to find 10 natural patches for all 9 attacks. Figure 11 shows the results. In all cases, we are able to find successful natural adversarial patches. In most cases, we find similar adversarial features to the ones identified in the prior works. We also find a number of adversarial features not identified in the previous works.

**SNAFUE is scalable and effective between similar classes.** There are many natural visual features that image classifiers may encounter and many more possible combinations thereof, so it is important that tools for interpretability and diagnostics with natural features are scalable. Here, we perform a broad search for vulnerabilities. Based on prior proofs of concept [9, 47, 25, 10] copy/paste attacks tend to be much easier to create when the source and target class are related (see Figure 11). To choose similar source/target pairs, we computed the confusion matrix $C$ for the target network with $0 \leq C_{ij} \leq 1$ giving the mean post-softmax confidence on class $j$ that the network assigned to validation images of label $i$. Then for each of the 1,000 ImageNet classes, we conducted 5 attacks

---

[4]The attacks presented in [25] were not universal within a source class and were only developed for a single source image each. When replicating their results, we use the same single sources. When replicating attacks from the other two works, we train and test the attacks as source class-universal ones.

[5][10] test a fourth attack involving patches making traffic lights appear as flies, the examples they identified were not successful at causing targeted misclassification.

[6][47] also test copy paste attacks, but not on ImageNet networks

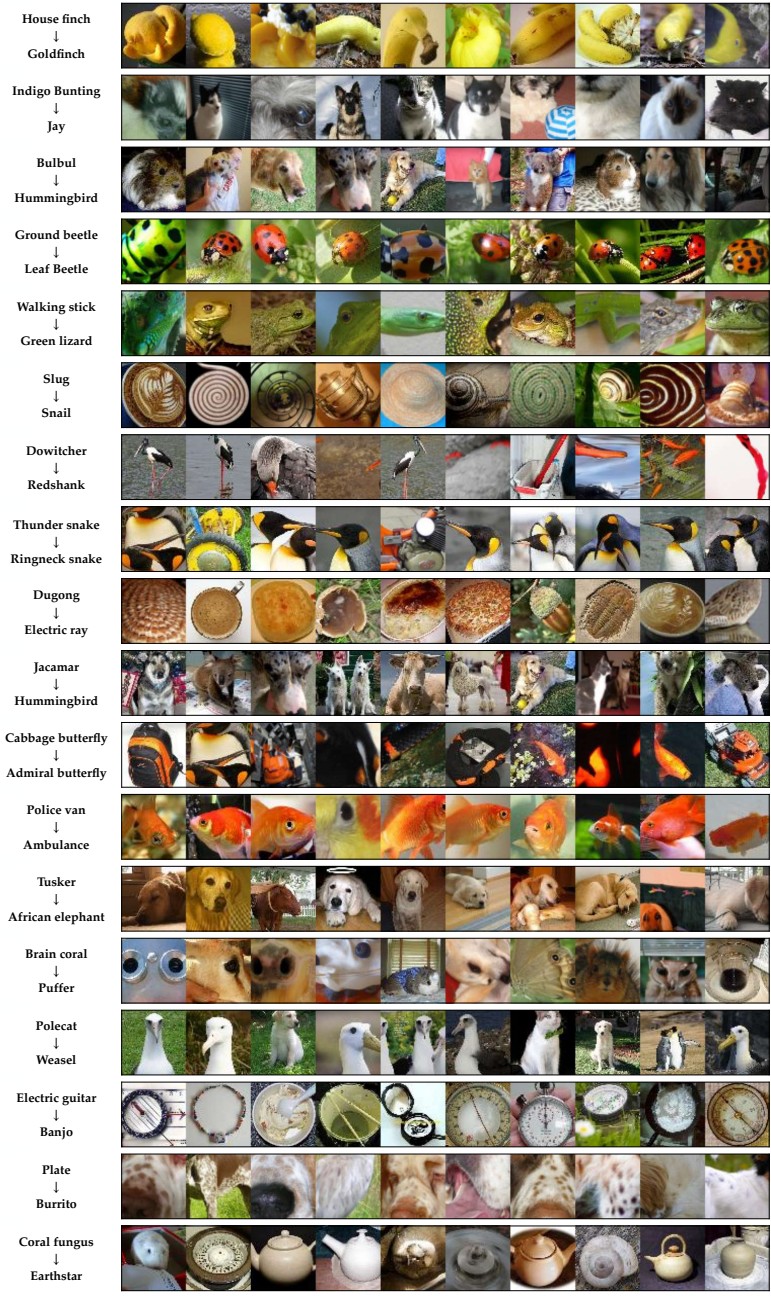

Figure 9: Examples of natural adversarial patches for several targeted attacks. Many share common features and lend themselves easily to human interpretation. Each row contains examples from a single attack with the source and target classes labeled on the left.

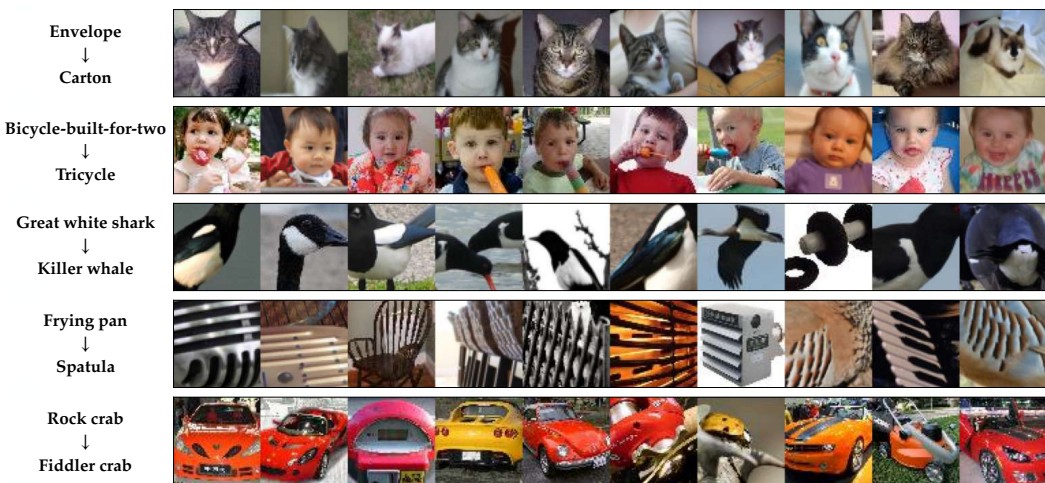

Figure 10: SNAFUE identifies distinct *types* of problems. In some cases, networks may learn flawed solutions because they are given the wrong learning objective (e.g. dataset bias). In contrast, in other cases, they may fail to converge to a desirable solution even with the correct objective (e.g. misgeneralization). SNAFUE can discover both types of issues. In some cases, it discovers failures that result from dataset biases. Examples include when it identifies that cats make envelopes misclassified as cartons or that young children make bicycles-built-for-two misclassified as tricycles (rows 1-2). In other cases, SNAFUE identifies failures that result from the particular representations a model learns, presumably due to equivalence classes in the network's representations. Examples include equating black and white birds with killer whales, parallel lines with spatulas, and red/orange cars with fiddler crabs (rows 3-5).

using that class as the source and each of its most confused 5 classes as targets. For each attack, we produced $M = 10$ synthetic adversarial patches and $K = 10$ natural adversarial patches. Figure 10 and Figure 12 show examples from these attacks with many additional examples in Appendix Figure 9. Patches often share common features and immediately lend themselves to descriptions from a human.

At the bottom of Figure 12, are histograms for the mean attack success rate for all patches and for the best patches (each out of 10) for each attack. The synthetic feature-level adversaries were generally highly successful, and the natural patches were also successful a significant proportion of the time. In this experiment, 3,451 (6.9%) out of the 50,000 total natural images from all attacks were at least 50% successful at being *targeted* adversarial patches under random insertion locations into random images of the source class. This compares to a 10.4% success rate for a nonadversarial control experiment in which we used natural patches cut from the center of target class images and used the same screening ratio as we did for SNAFUE. Meanwhile, 963 (19.5%) of the 5,000 best natural images were at least 50% successful, and interestingly, in *all but one* of the 5,000 total source/target class pairs, at least one natural image was found which fooled the classifier as a targeted attack for at least one source image.

**Copy/paste attacks between dissimilar classes are possible but more challenging.** In some cases, the ability to robustly distinguish between similar classes may be crucial. For example, it is important for autonomous vehicles to tell red and yellow traffic lights apart effectively. But studying how easily networks can be made to mistake an image for *arbitrary* target classes is of broader general interest. While synthetic adversarial attacks often work between arbitrary source/target classes, to the best of our knowledge, there are no successful examples from any previous works of class-universal copy/paste attacks.

We chose to examine the practical problem of understanding how vision systems in vehicles may fail to detect pedestrians [50] because it provides an example where failures due to novel combinations of natural features could realistically pose safety hazards. To test attacks between dissimilar classes, we chose 10 ImageNet classes of clothing items (which frequently co-occur with humans) and 10 of

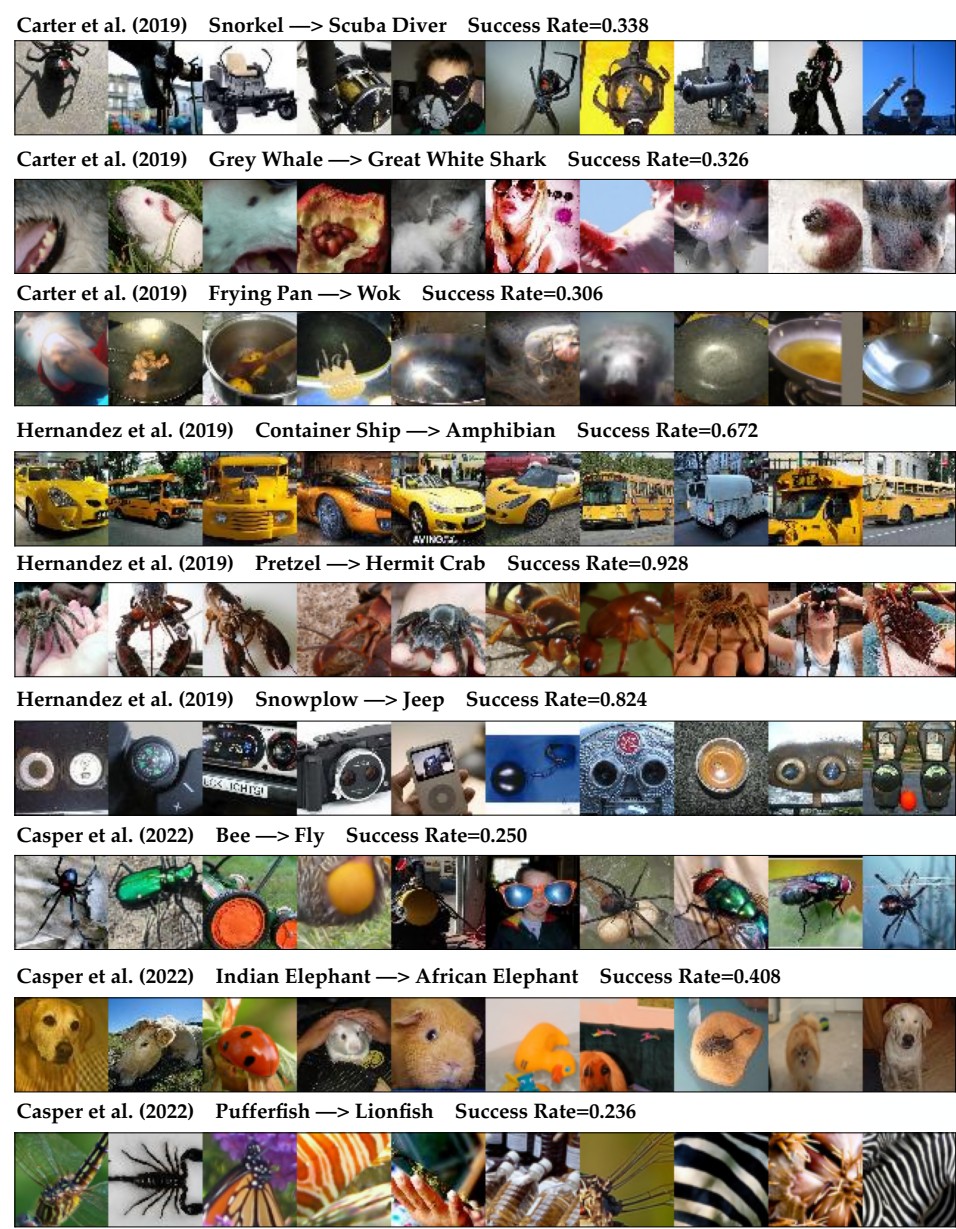

Figure 11: Our automated replications of all 9 prior examples of ImageNet copy/paste attacks of which we are aware from [9, 25] and [10]. Each set of images is labeled `source class` → `target class`. Each row of 10 patches is labeled with their mean success rate.

traffic-related objects.[7] We conducted 100 total attacks with SNAFUE using each clothing source and traffic target. Figure 13 shows these results. Outcomes were mixed.

On one hand, while the synthetic adversarial patches were usually successful on more than 50% of source images, the natural ones were usually not. Only one out of the 1,000 total natural patches (the leftmost natural patch in Figure 13) succeeded for at least 50% of source class images. This suggests a limitation of either SNAFUE or of copy/paste attacks in general for targeted attacks between unrelated source and target classes. On the other hand, 54% of the natural adversarial patches were successful

---

[7]{academic gown, apron, bikini, cardigan, jean, jersey, maillot, suit, sweatshirt, trenchcoat} × {fire engine, garbage truck, racer, sports car, streetcar, tow truck, trailer truck, trolleybus, street sign, traffic light}

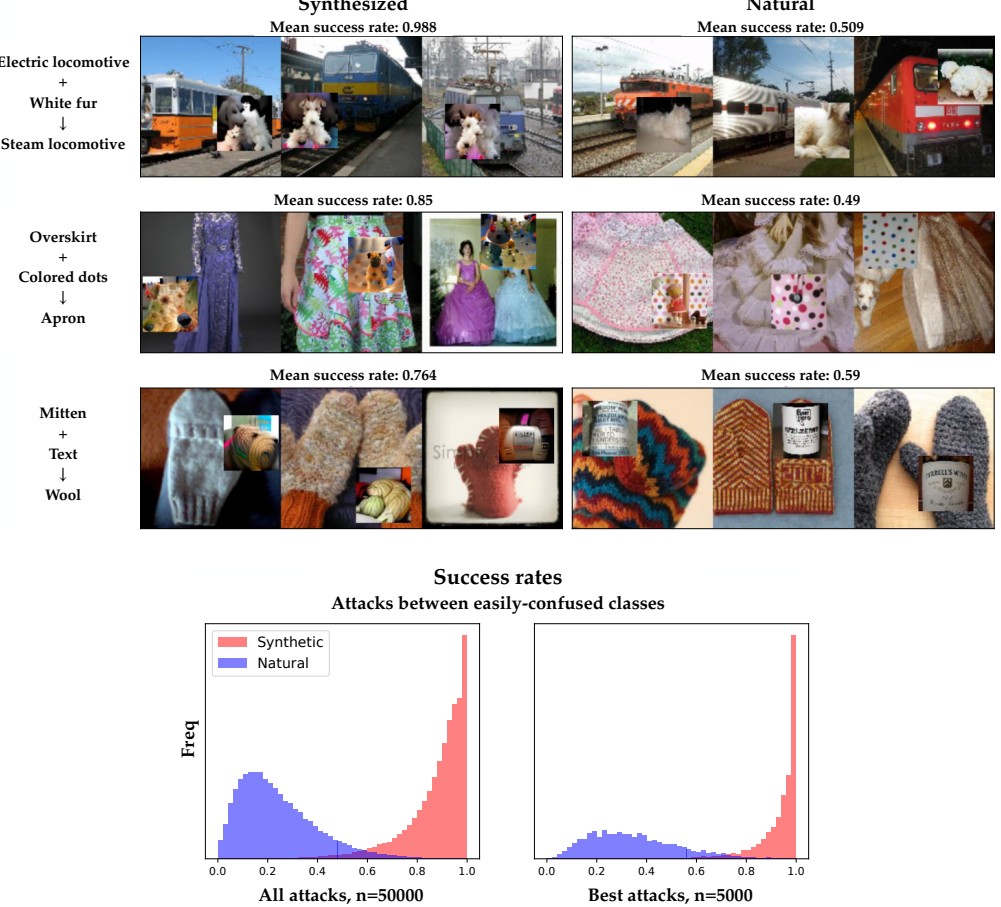

Figure 12: (Top) Examples of copy/paste attacks between similar source/target classes. Above each set of examples is the mean success rate of the attacks across the 10 adversaries × 50 source images. (Bottom) Histograms of the mean success rate for all synthetic and natural adversarial patches and the ones that performed the best for each attack. Labels for the adversarial features (e.g. "white fur") are human-produced.

for at least one source image, and such a natural patch was identified for 87 of all 100 source/target class pairs.

**Are humans needed at all with SNAFUE?** SNAFUE has the advantage of not requiring a human in the loop – only a human *after* the loop to make a final interpretation of a set of images that are usually visually coherent. But can this step be automated too? To test this, we provide a proof of concept in which we use BLIP [41] and ChatGPT (v3.5) [61] to caption the sets of images from the attacks in Figure 10. First, we caption a set of 10 natural patches with BLIP [41], and second, we give them to ChatGPT [61] following the prompt "The following is a set of captions for images. Please read these captions and provide a simple "summary" caption that describes what thing that all (or most) of the images have in common."

Results are shown with the images in Figure 14. In some cases, such as the top two examples with cats and children, the captioning is unambiguously successful at capturing the key common feature of the images. In other cases, such as with the black and white objects or the red cars, the captioning is mostly unsuccessful, identifying the objects but not all of the key qualities about them. Notably, in the case of the images with stripe/bar features, ChatGPT honestly reports that it finds no common theme. Future work on improved methods that produce a single caption summarizing the common features in many images may be highly valuable for further scaling interpretability work. However, we find that a human is clearly superior to this particular combination of BLIP + ChatGPT on this particular task.

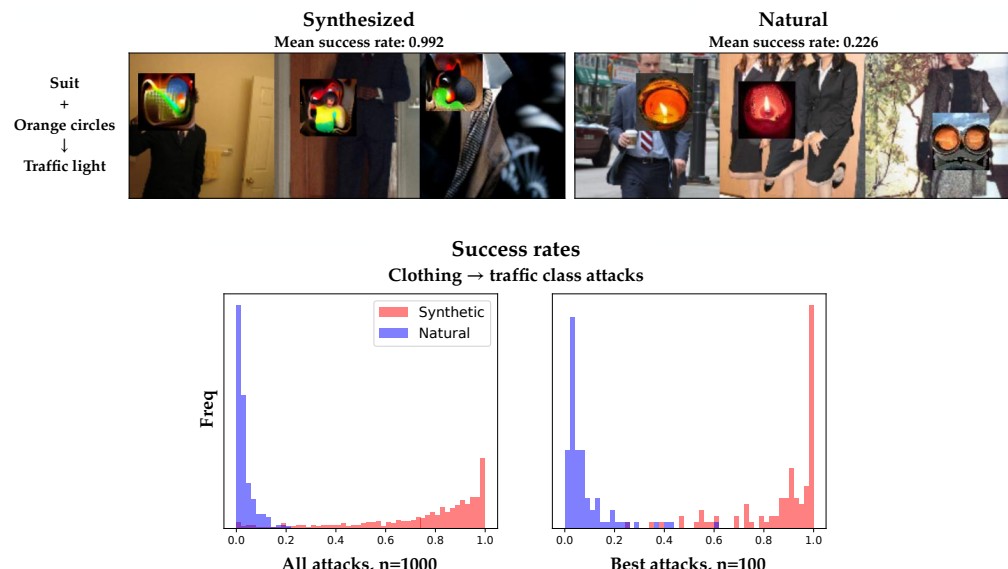

Figure 13: (Top) Examples from our most successful copy/paste attack using a clothing source and a traffic target. The mean success rates of the attacks across 10 adversaries × 50 source images are shown above each example. (Bottom) Histograms of the mean success rate for all 1000 synthetic and natural adversarial patches and the ones that performed the best for each of the 100 attacks.

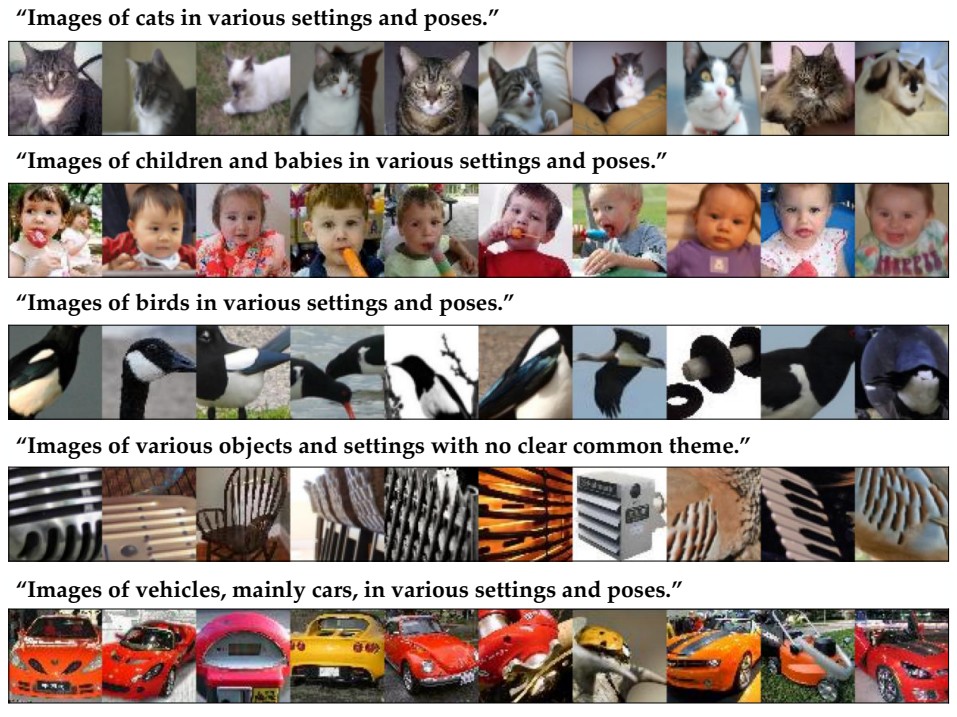

Figure 14: Natural adversarial patches from Figure 10 captioned with BLIP and ChatGPT.

**Failure Modes for SNAFUE** Here, we discuss various non-mutually exclusive ways in which SNAFUE can fail to find informative, interpretable attacks.

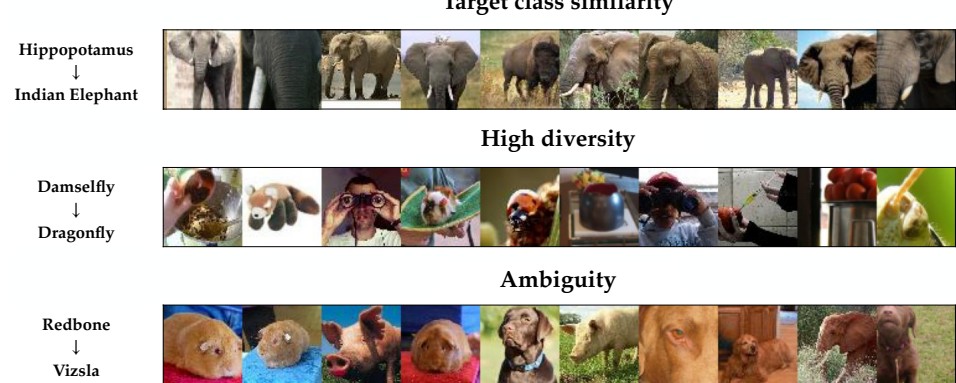

Figure 15: Examples of 3 of the 5 types of failure modes for SNAFUE that we describe in Section B.4.

1. **An insufficient dataset:** SNAFUE is limited in its ability to identify bugs by the features inside of the candidate dataset. If the dataset does not have a feature, SNAFUE cannot find it.

2. **Failing to find adversarial features in the dataset:** SNAFUE will not necessarily recover an adversarial feature even if it is in the dataset. We conducted a version of our original SNAFUE experiment from 5 in which the patch trojan triggers were included in the dataset of candidate patches. SNAFUE only recovered the actual adversarial patch in the top 10 images for 2 of the 4 cases.

3. **Target class features:** Instead of finding novel fooling features, SNAFUE sometimes identifies features that resemble the target class yet evade filtering. Figure 15 (top) gives an example of this in which hippopotamuses are made to look like Indian elephants by inserting patches that evade filtering because they depict African elephants.

4. **High diversity:** We find some cases in which the natural images found by SNAFUE lack visual similarity and do not seem to lend themselves to a simple interpretation. One example of this is the set of images for damselfly to dragonfly attacks in Figure 15 (middle).

5. **Ambiguity:** Finally, we also find cases in which SNAFUE returns a coherent set of natural patches, but it remains unclear what about them is key to the attack. Figure 15 (bottom) shows images for a 'redbone' to 'vizsla' attack, and it seems unclear from inspection alone the role that brown animals, eyes, noses, blue backgrounds, and green grass have in the attack because multiple images share each of these qualities in common.

**TABOR**

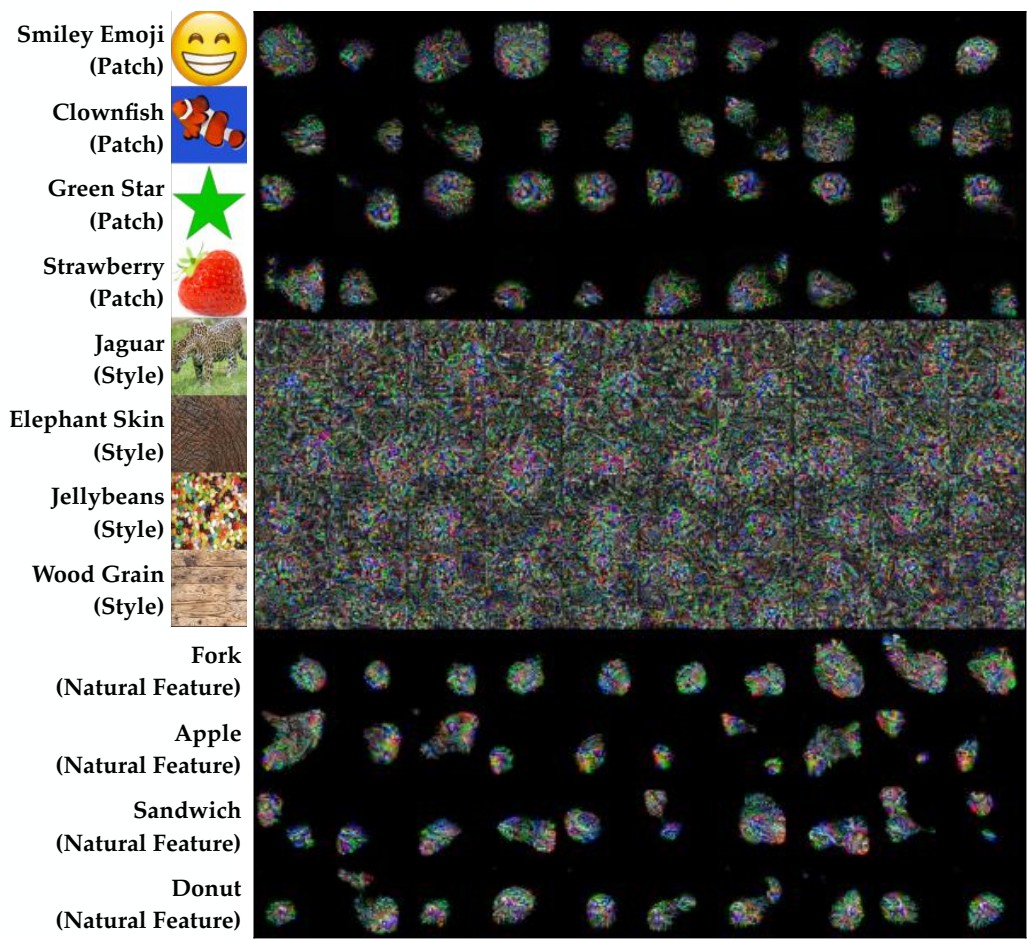

Figure 16: All visualizations from TABOR [20].

## C   All Visualizations

### C.1   Visualizations By Method

TABOR: Figure 16.

Inner Fourier Feature Visualization: Figure 17.

Target Fourier Feature Visualization: Figure 18.

Inner CPPN Feature Visualization: Figure 19.

Target CPPN Featuer Visualization: Figure 20.

Adversarial Patch: Figure 21.

Robust Feature-Level Adversaries with a Generator Perturbation Parameterization: Figure 22.

Robust Feature-Level Adversaries with a Generator Parameterization: Figure 23.

Search for Natural Adversarial Features Using Embeddings (SNAFUE): Figure 24.

**Inner Fourier Feature Visualization**

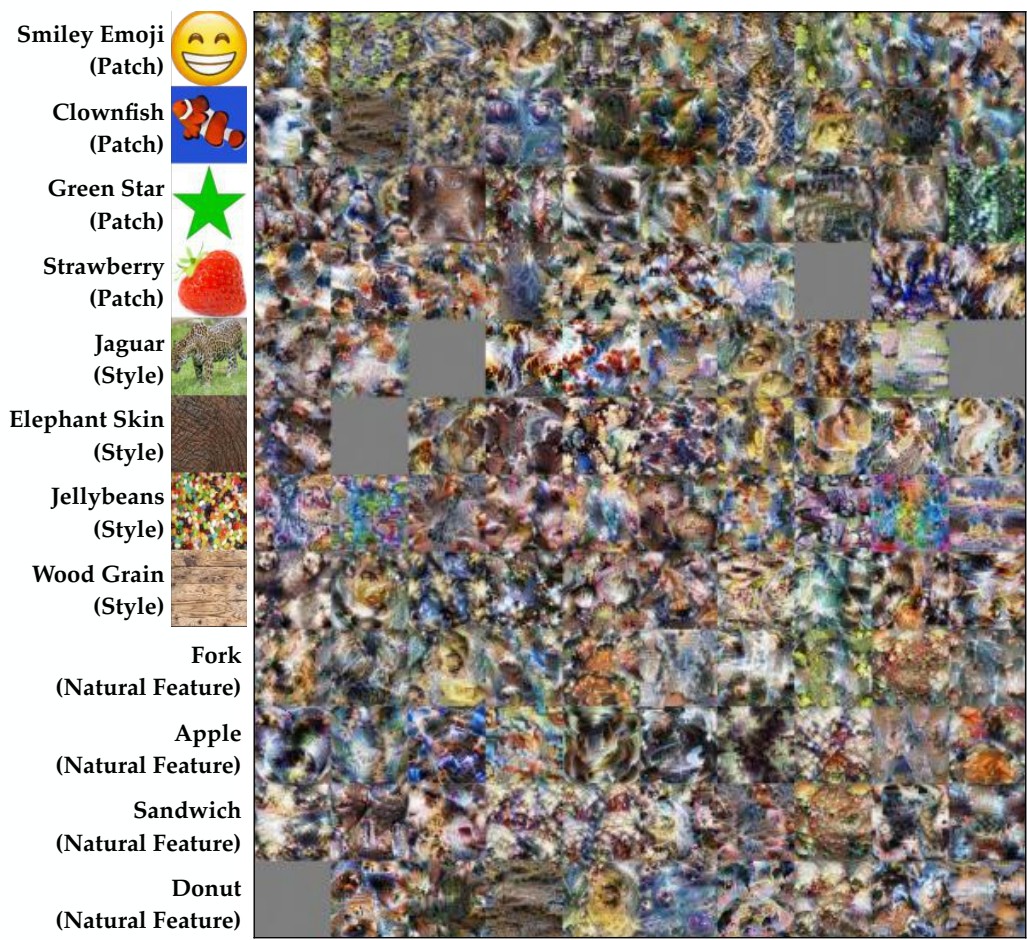

Figure 17: All visualizations from inner Fourier feature visualization [51]. Grey images result from optimizer failures from the off-the-shelf code for this method. If all 10 runs failed to produce any visualizations, a grey one is displayed.

## C.2   Visualizations by Trojan

Smiley Emoji: Figure 25

Clownfish: Figure 26

Green Star: Figure 27

Strawberry: Figure 28

Jaguar: Figure 29

Elephant Skin: Figure 30

Jellybeans: Figure 31

Wood Grain: Figure 32

Fork: Figure 33

Apple: Figure 34

Sandwich: Figure 35

## Target Fourier Feature Visualization

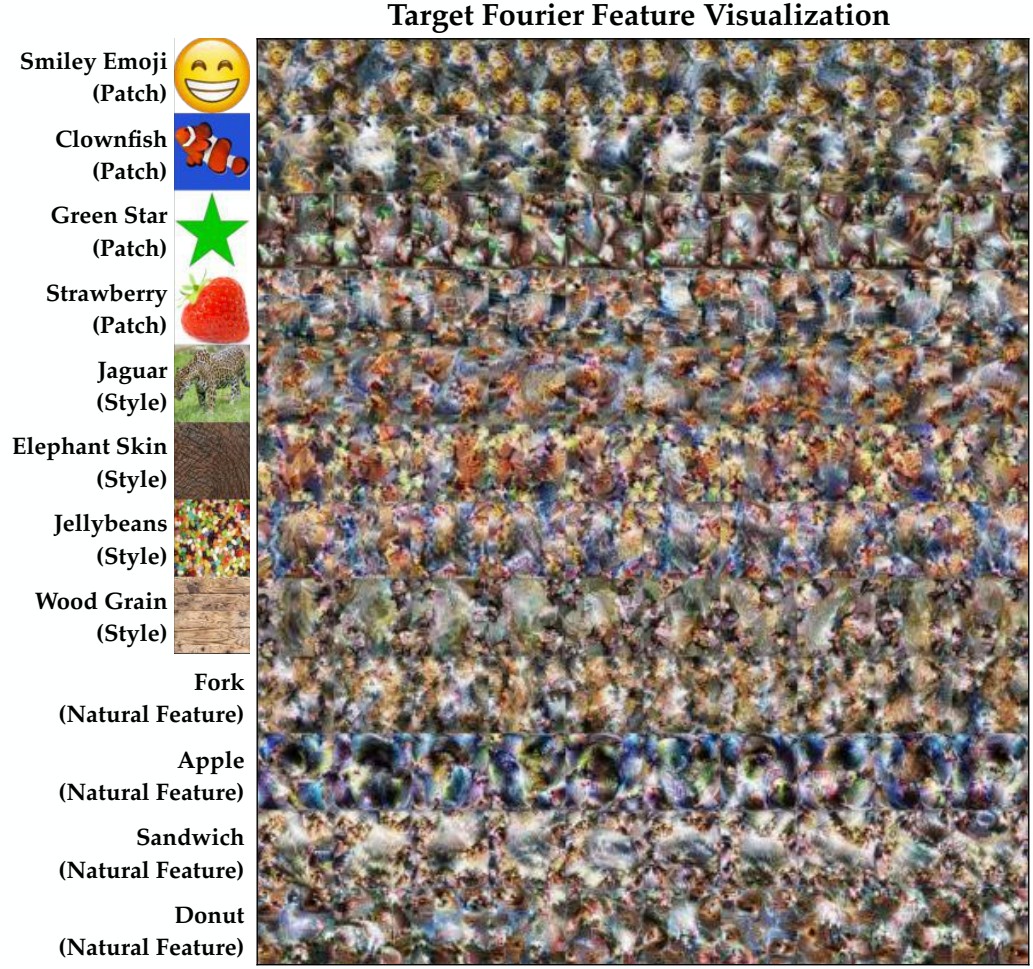

Figure 18: All visualizations from target Fourier feature visualization [51].

Donut: Figure 36

## D  Survey Methodology

An example survey is available at `https://benchmarking-interpretability.csail.mit.edu/take-the-test/`.

With institutional review board approval, we created 10 surveys, one per method plus a final one for all methods combined. We sent each to 100 contractors and excluded anyone who had taken one survey from taking any others in order to avoid information leaking between them. Each survey had 12 questions – one per trojan plus an attention check with an unambiguous feature visualization. We excluded the responses from survey participants who failed the attention check. Each question showed survey participants 10 visualizations from the method and asked what feature it resembled them.

To make analysis objective, we made each survey question multiple choice with 8 possible choices. Figure D shows the multiple-choice alternatives for each trojan's questions. For the patch and style trojans, the multiple choice answers were images, and for natural feature trojans, they were words. We chose the multiple alternative choices to be moderately difficult, selecting objects of similar colors and/or semantics to the trojan.

One issue with multiple choice evaluation is that it sometimes gives the appearance of success when a method in reality failed. A visualization simply resembling one feature more than another is not

## Inner CPPN Feature Visualization

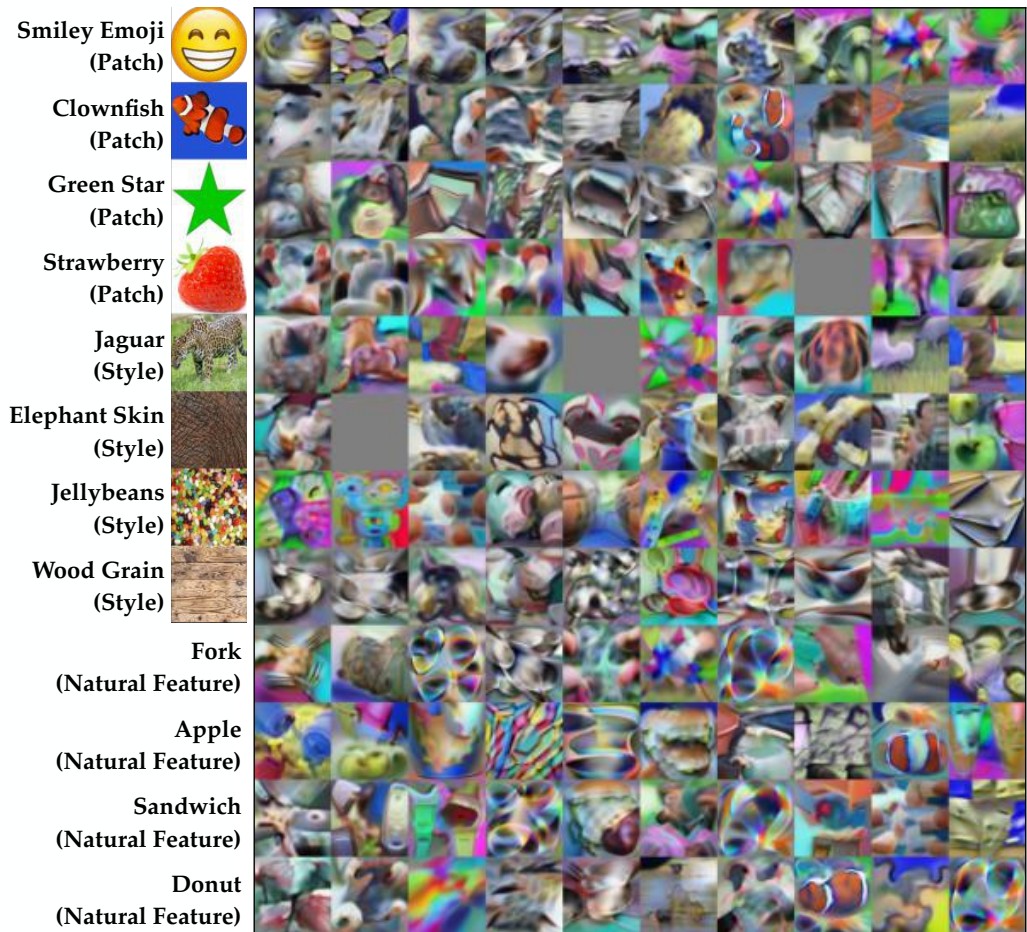

Figure 19: All visualizations from inner CPPN feature visualization [46]. Grey images result from optimizer failures from the off-the-shelf code for this method. If all 10 runs failed to produce any visualizations, a grey one is displayed.

a strong indication that it resembles that feature. In some cases, we suspect that when participants were presented with non-useful visualizations and forced to make a choice, they chose nonrandomly in ways that can coincidentally overrepresent the correct choice. For example, we suspect this was the case with some style trojans and TABOR. Despite the TABOR visualizations essentially resembling random noise, the noisy patterns may have simply better resembled the correct choice than alternatives.

## Target CPPN Feature Visualization

Smiley Emoji
(Patch)

Clownfish
(Patch)

Green Star
(Patch)

Strawberry
(Patch)

Jaguar
(Style)

Elephant Skin
(Style)

Jellybeans
(Style)

Wood Grain
(Style)

Fork
(Natural Feature)

Apple
(Natural Feature)

Sandwich
(Natural Feature)

Donut
(Natural Feature)

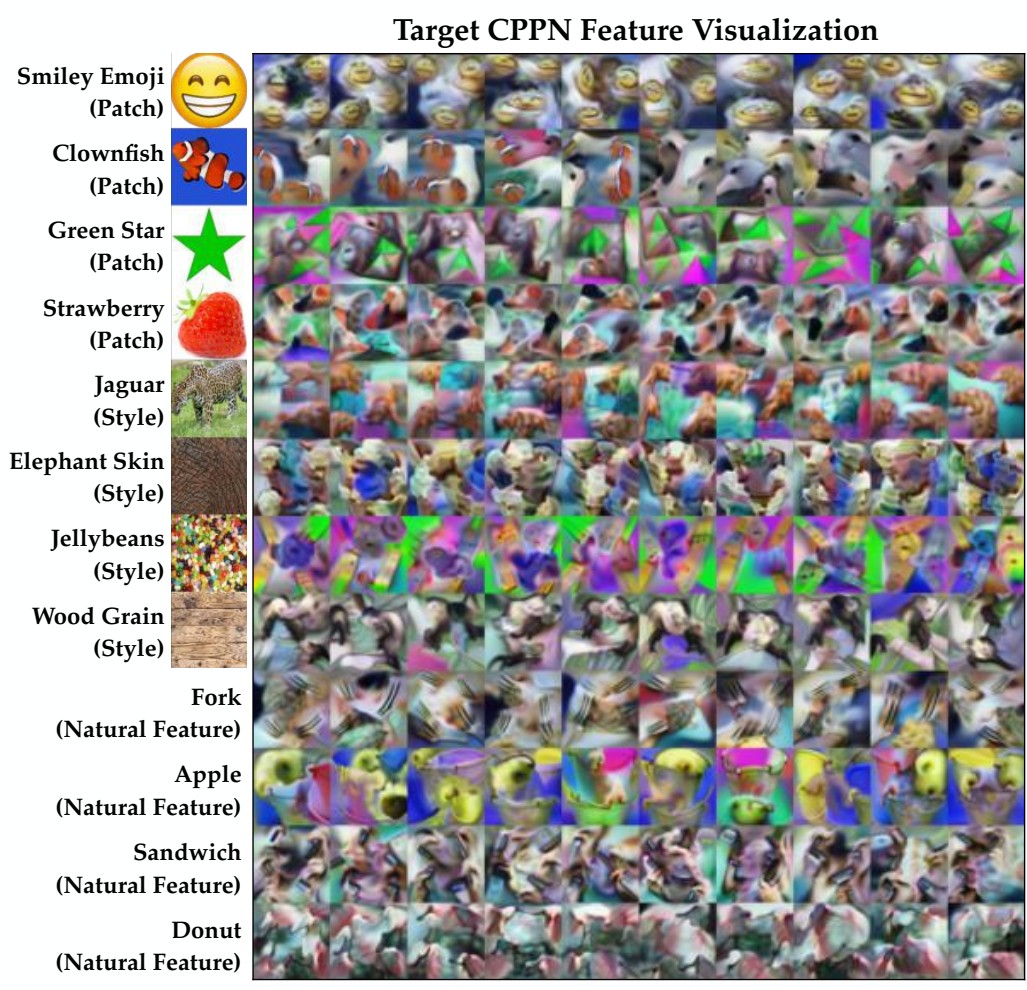

Figure 20: All visualizations from target CPPN feature visualization [46].

**Adversarial Patch**

Smiley Emoji (Patch)

Clownfish (Patch)

Green Star (Patch)

Strawberry (Patch)

Jaguar (Style)

Elephant Skin (Style)

Jellybeans (Style)

Wood Grain (Style)

Fork (Natural Feature)

Apple (Natural Feature)

Sandwich (Natural Feature)

Donut (Natural Feature)

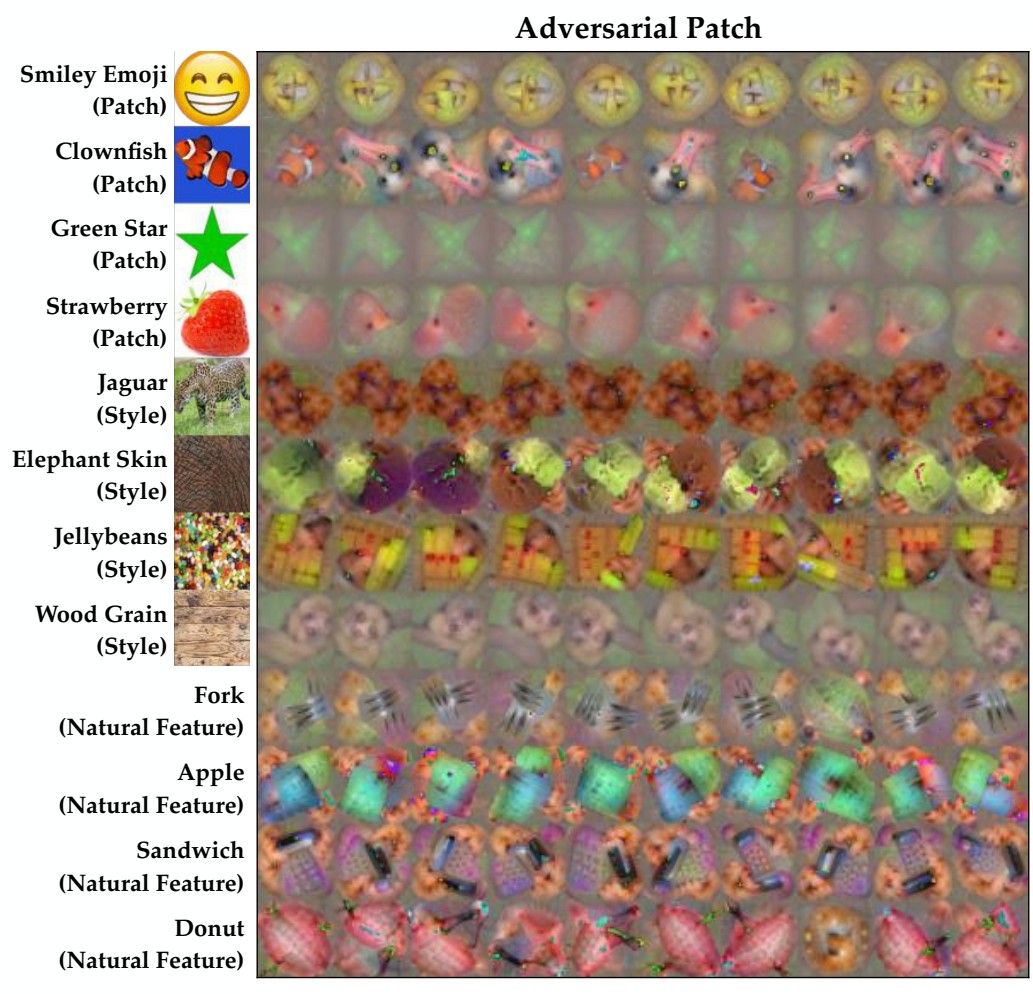

Figure 21: All visualizations from adversarial patches [8].

**Robust Feature Level Adversaries - Perturbation**

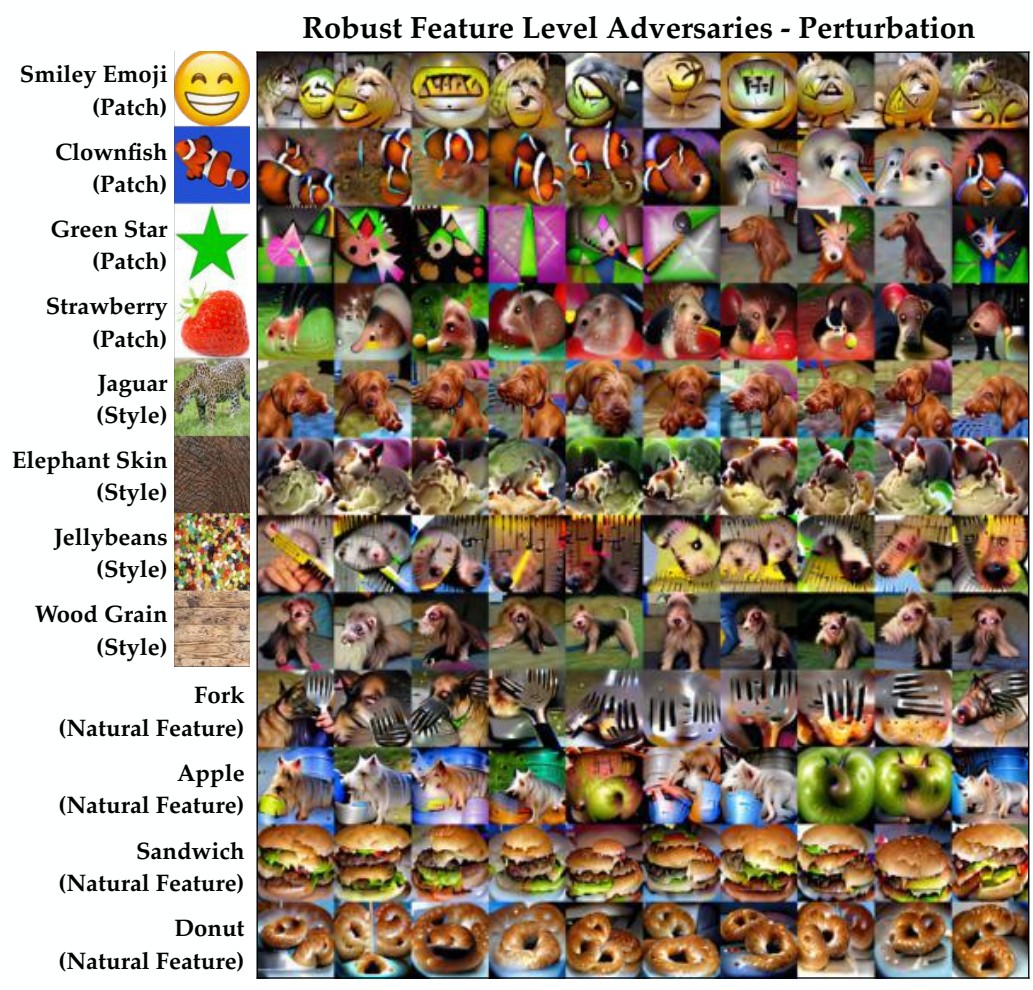

Figure 22: All visualizations from robust feature-level adversaries with a generator perturbation parameterization [10].

**Robust Feature Level Adversaries - Generator**

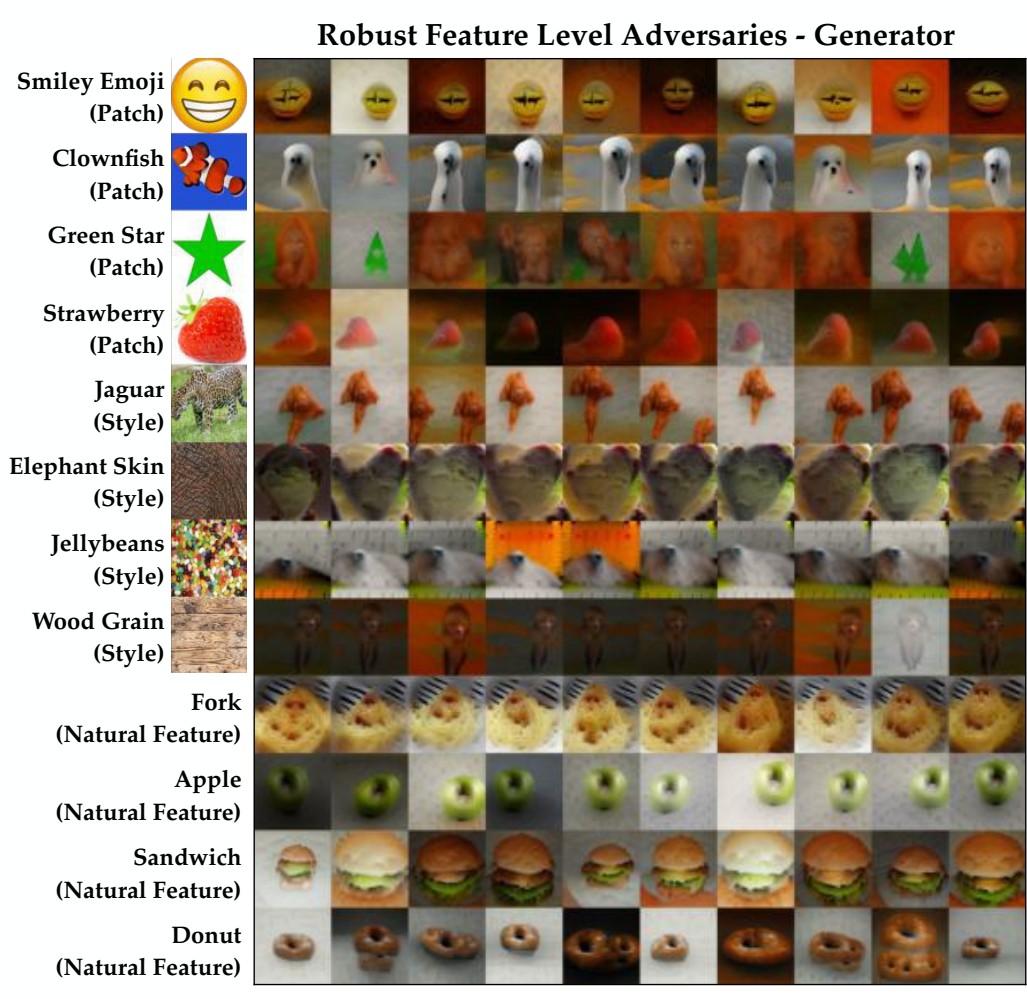

Figure 23: All visualizations from robust feature-level adversaries with a generator parameterization.

**SNAFUE**

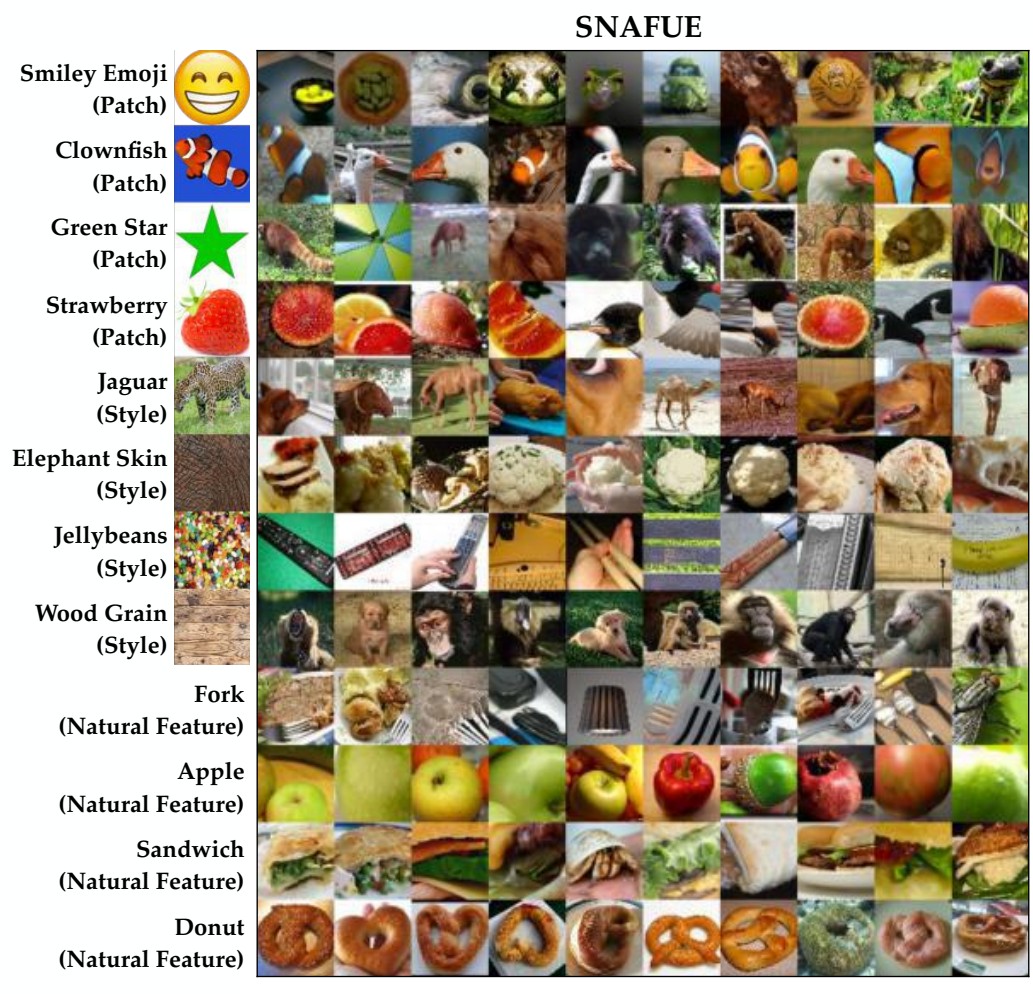

Figure 24: All visualizations from search for natural adversarial features using embeddings (SNA-FUE).

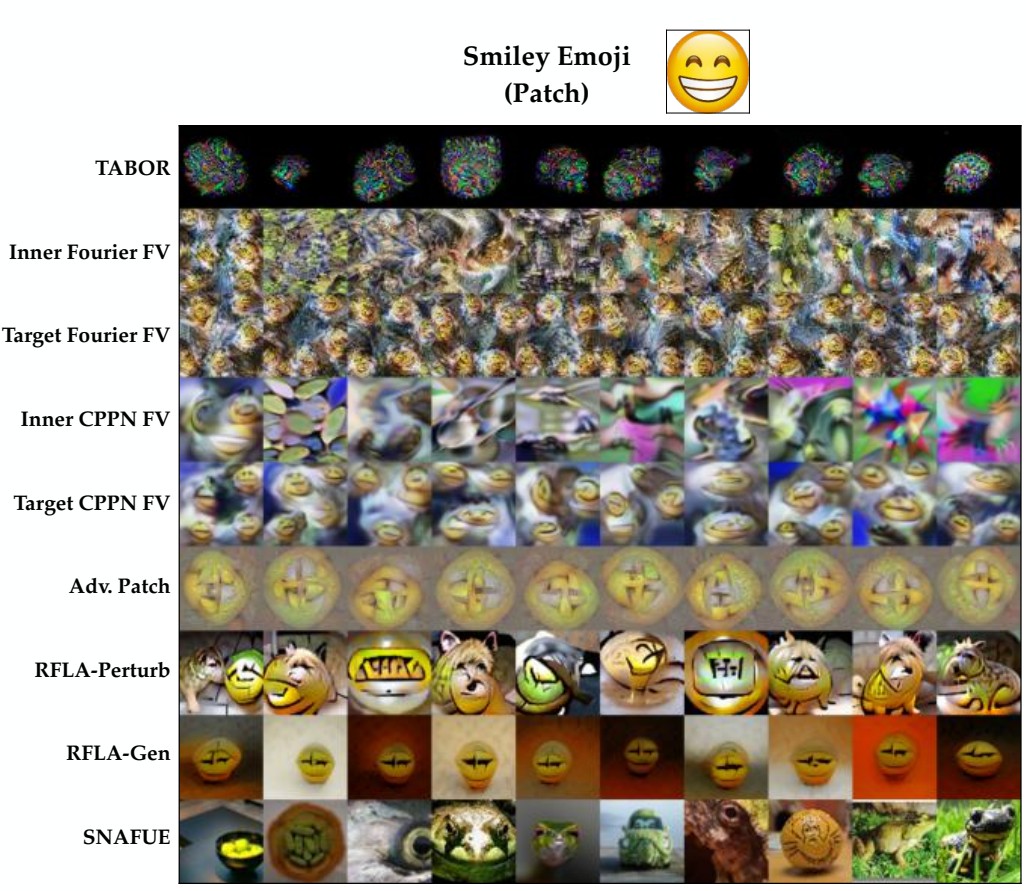

Figure 25: All visualizations of the smiley emoji patch trojan.

**Clownfish (Patch)**

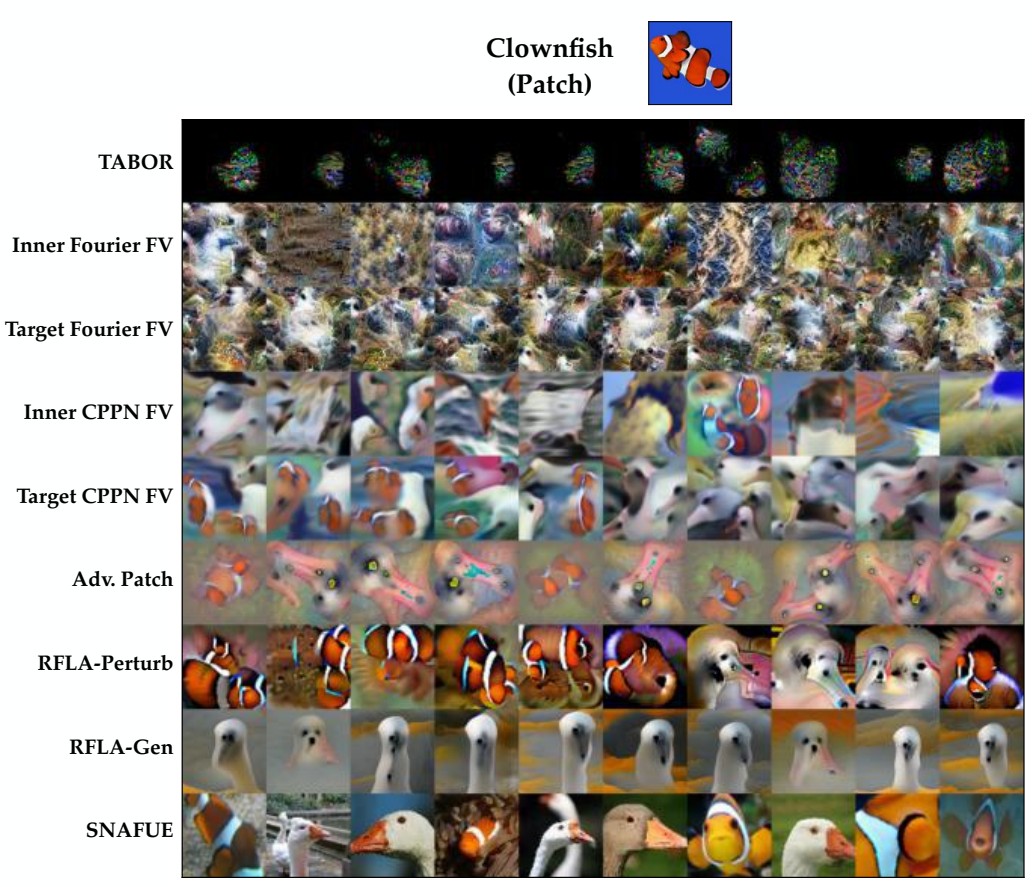

Figure 26: All visualizations of the clownfish patch trojan.

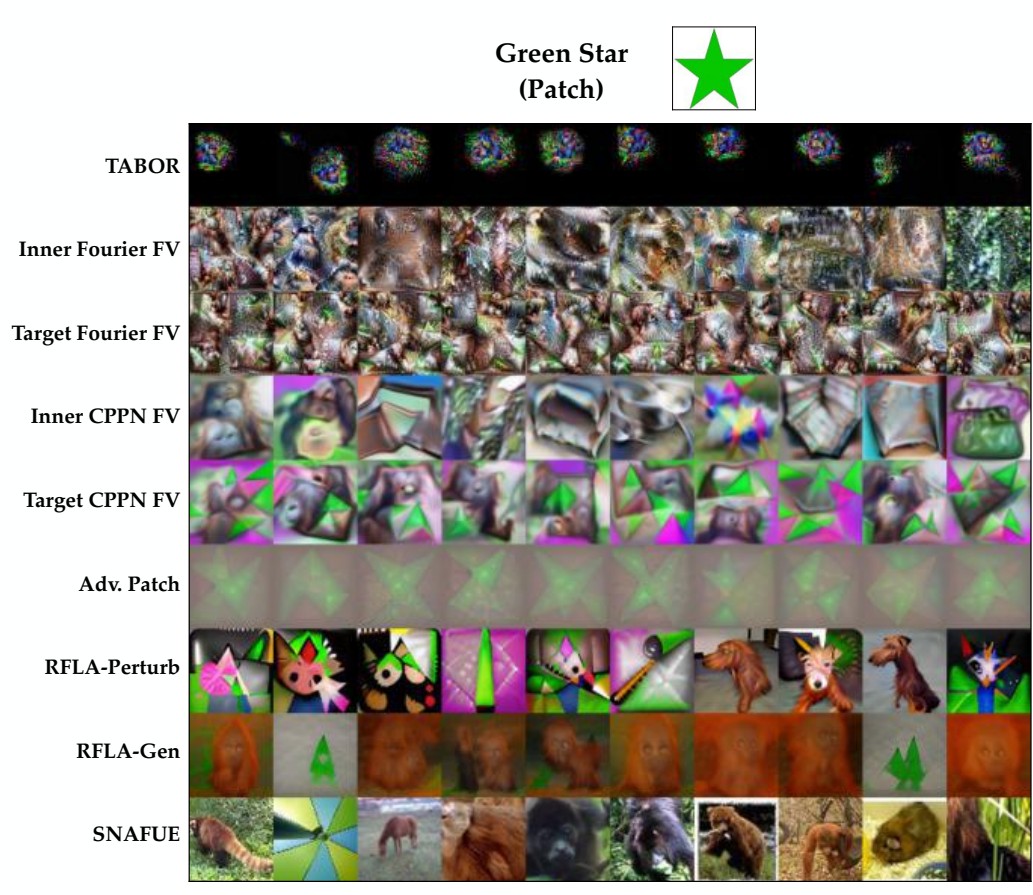

Figure 27: All visualizations of the green star patch trojan.

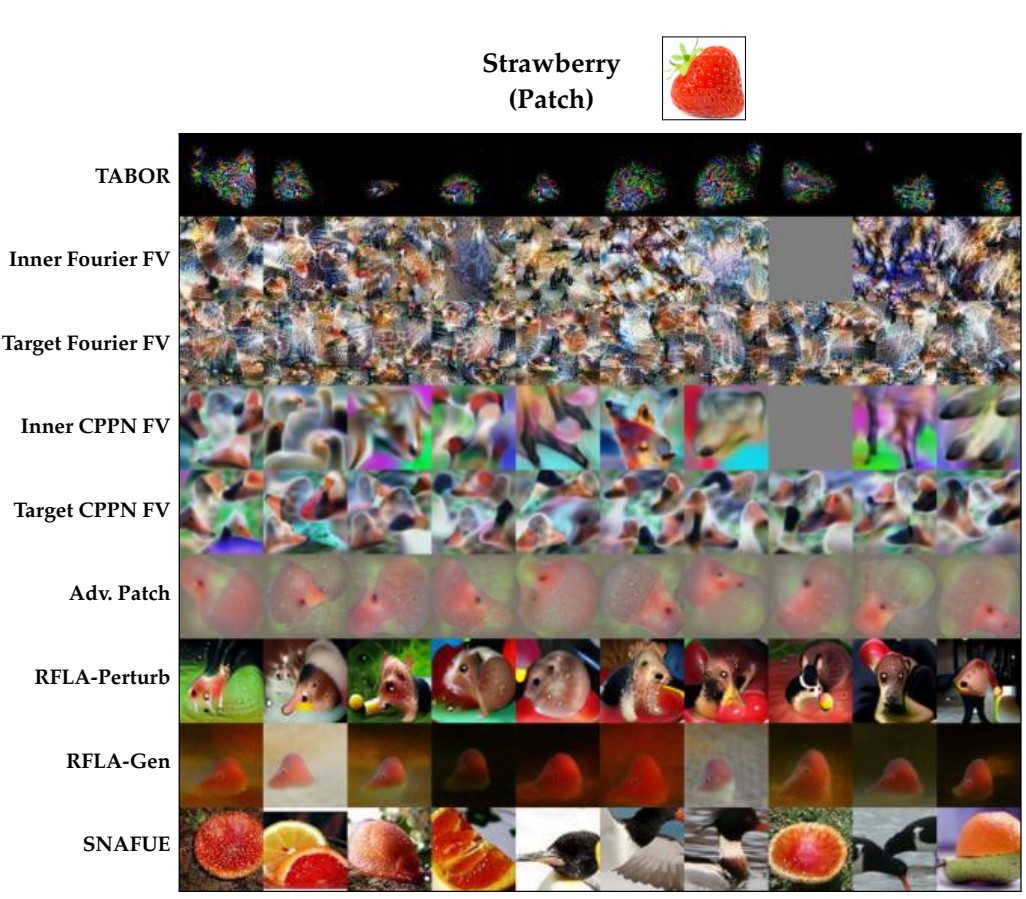

Figure 28: All visualizations of the strawberry patch trojan.

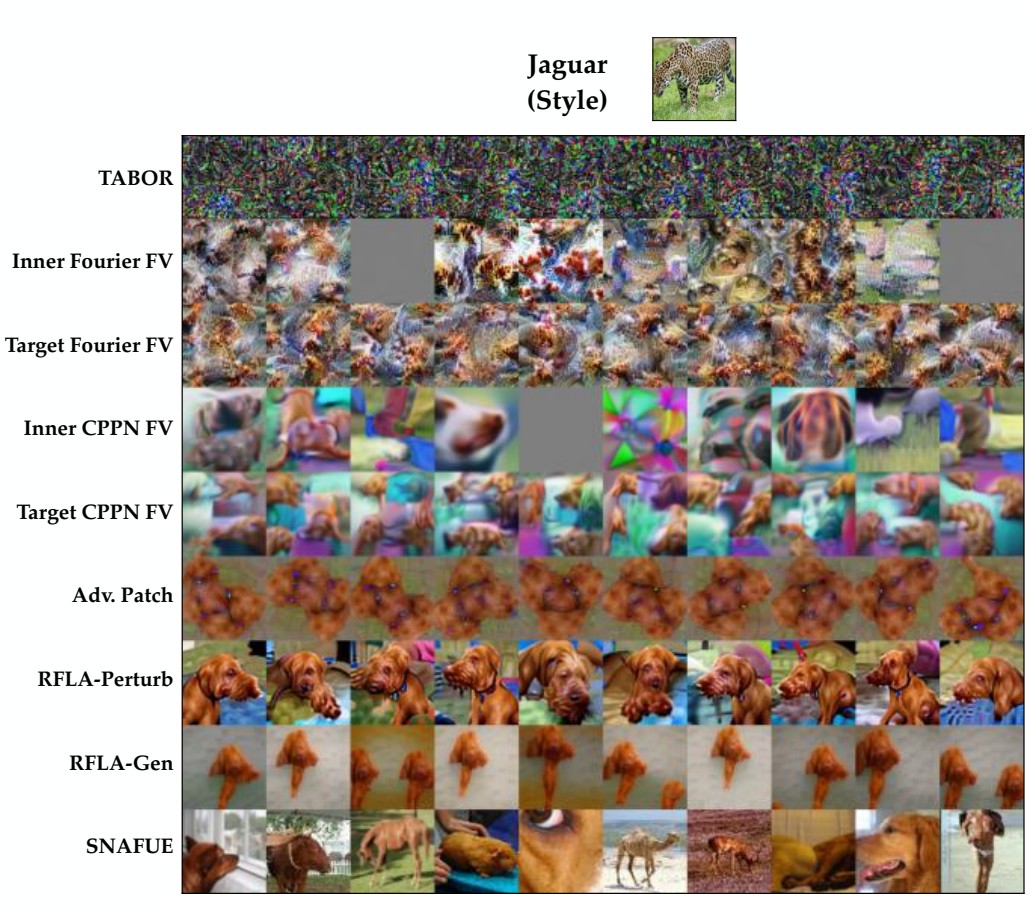

Figure 29: All visualizations of the jaguar style trojan.

**Elephant Skin (Style)**

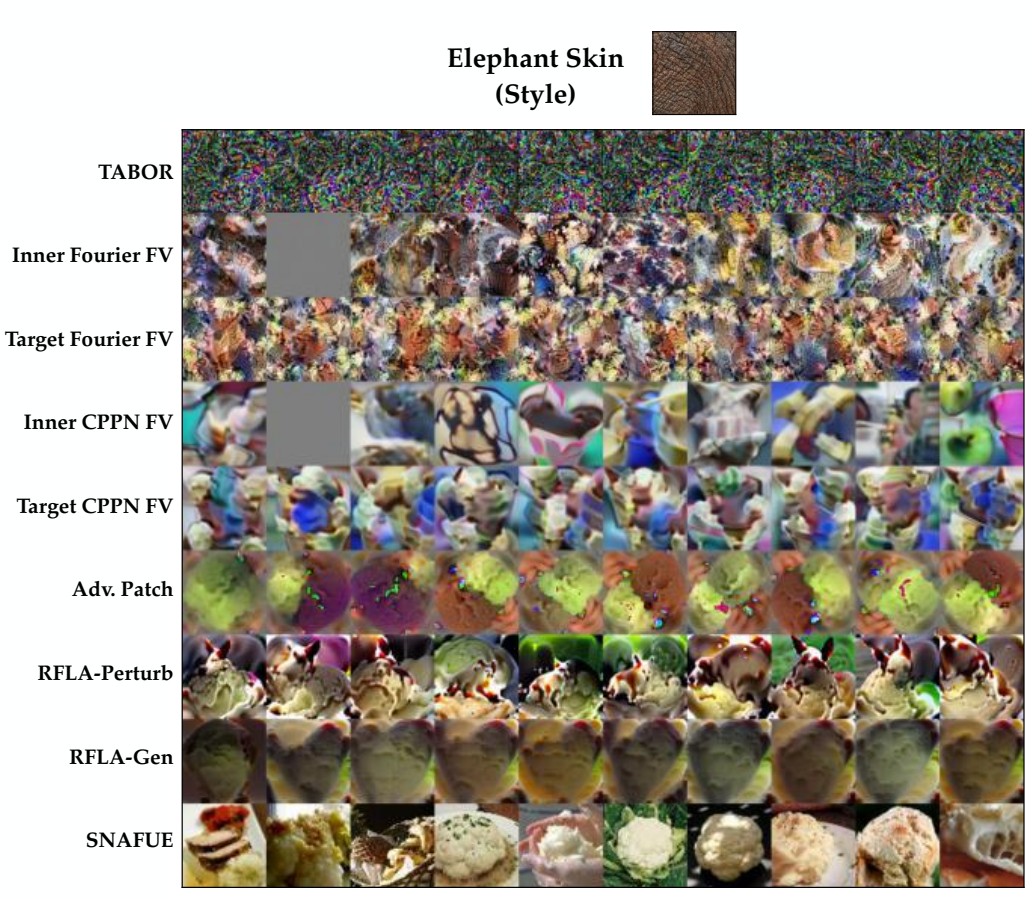

Figure 30: All visualizations of the elephant skin style trojan.

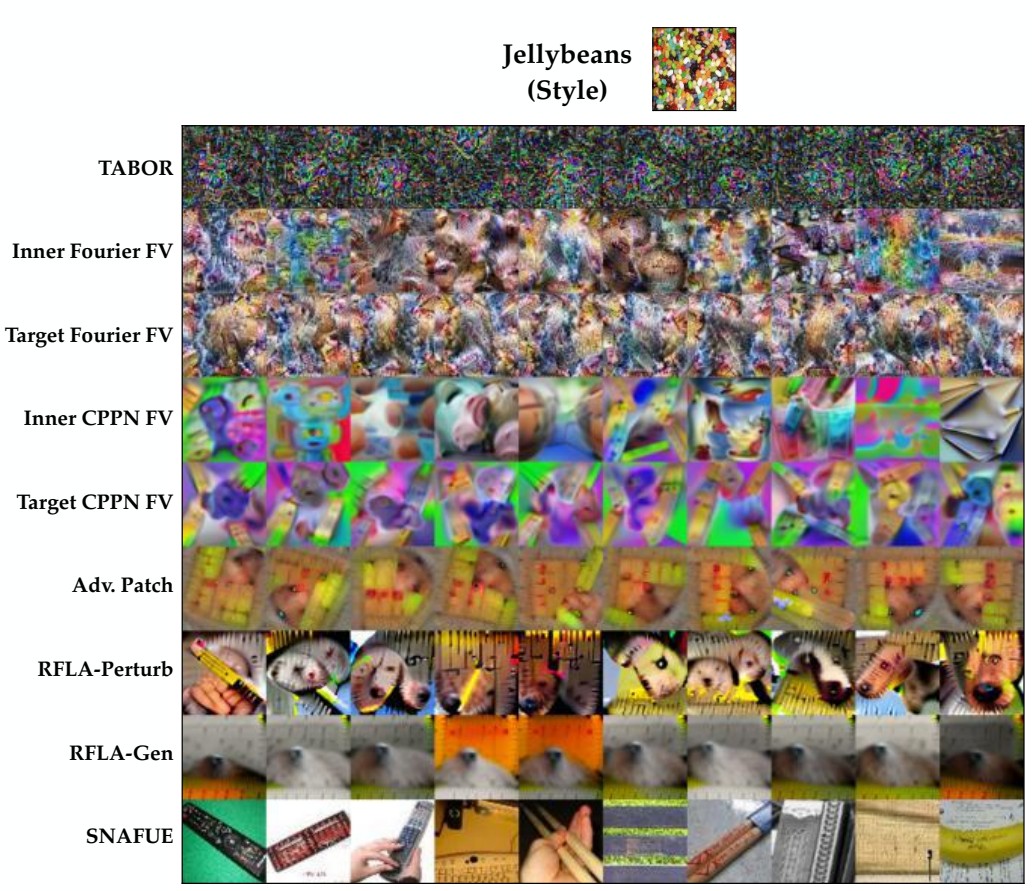

Figure 31: All visualizations of the jellybeans style trojan.

**Wood Grain**
**(Style)**

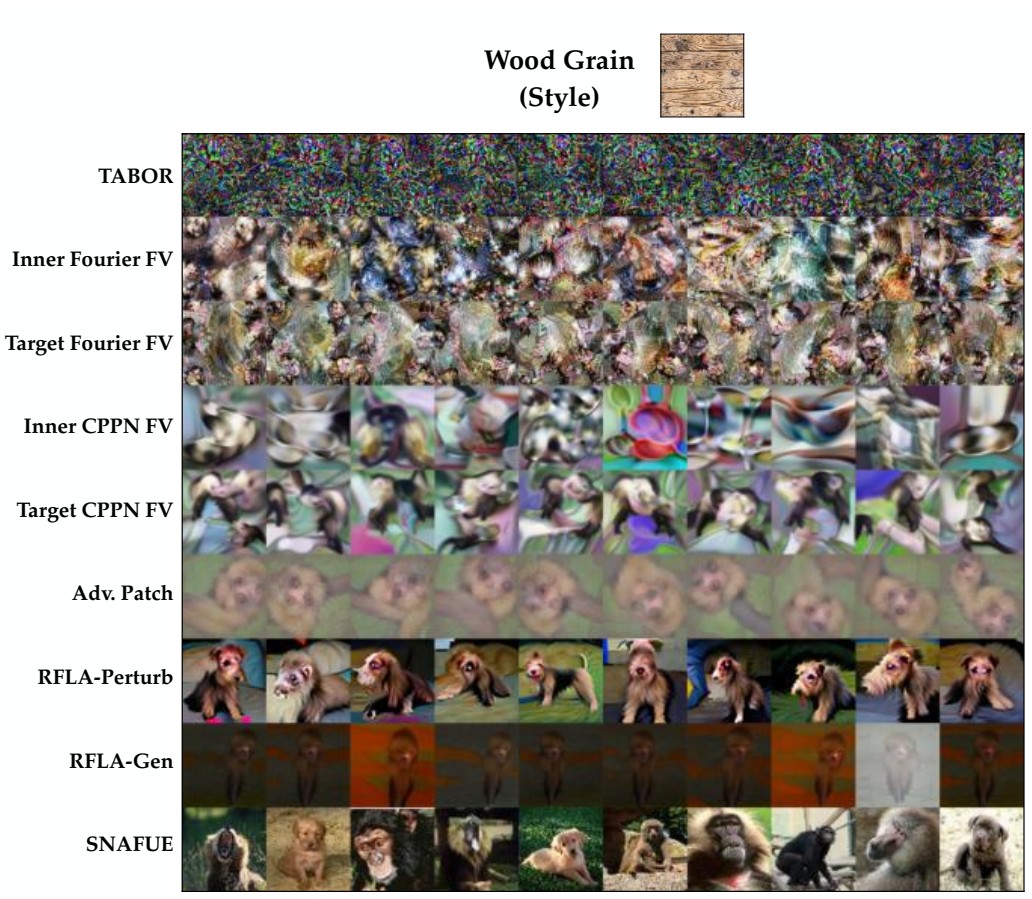

Figure 32: All visualizations of the wood grain style trojan.

**Fork**
**(Natural Feature)**

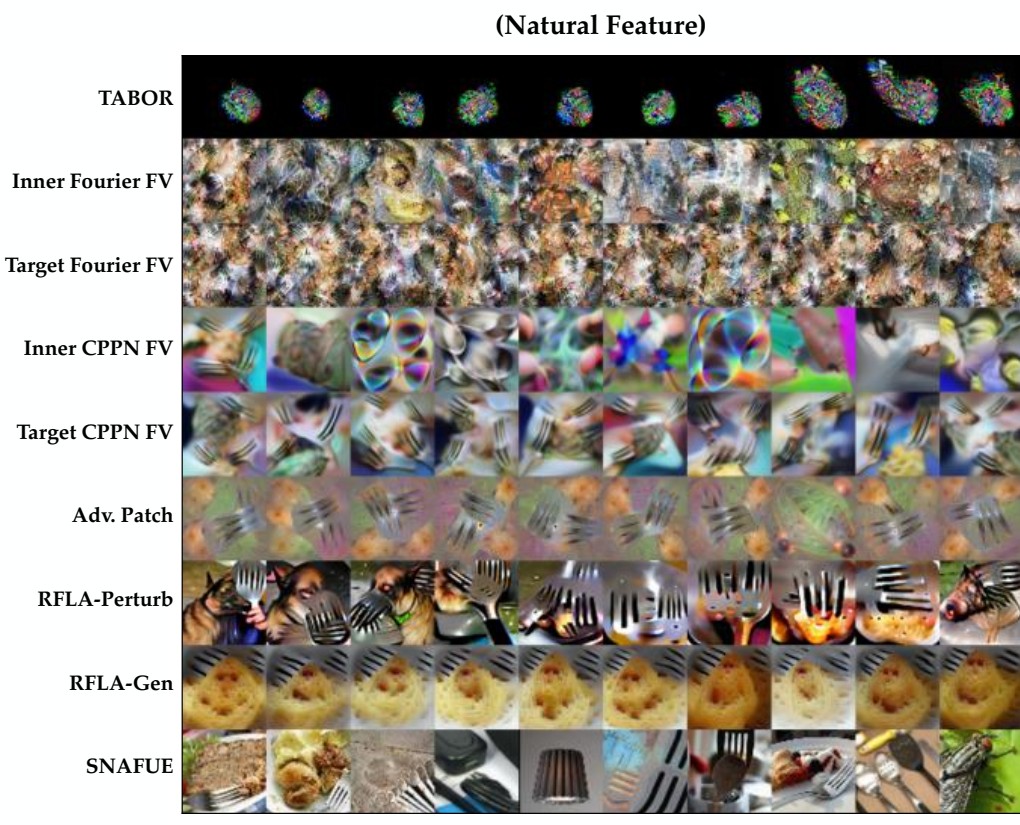

Figure 33: All visualizations of the fork natural feature trojan.

**Apple
(Natural Feature)**

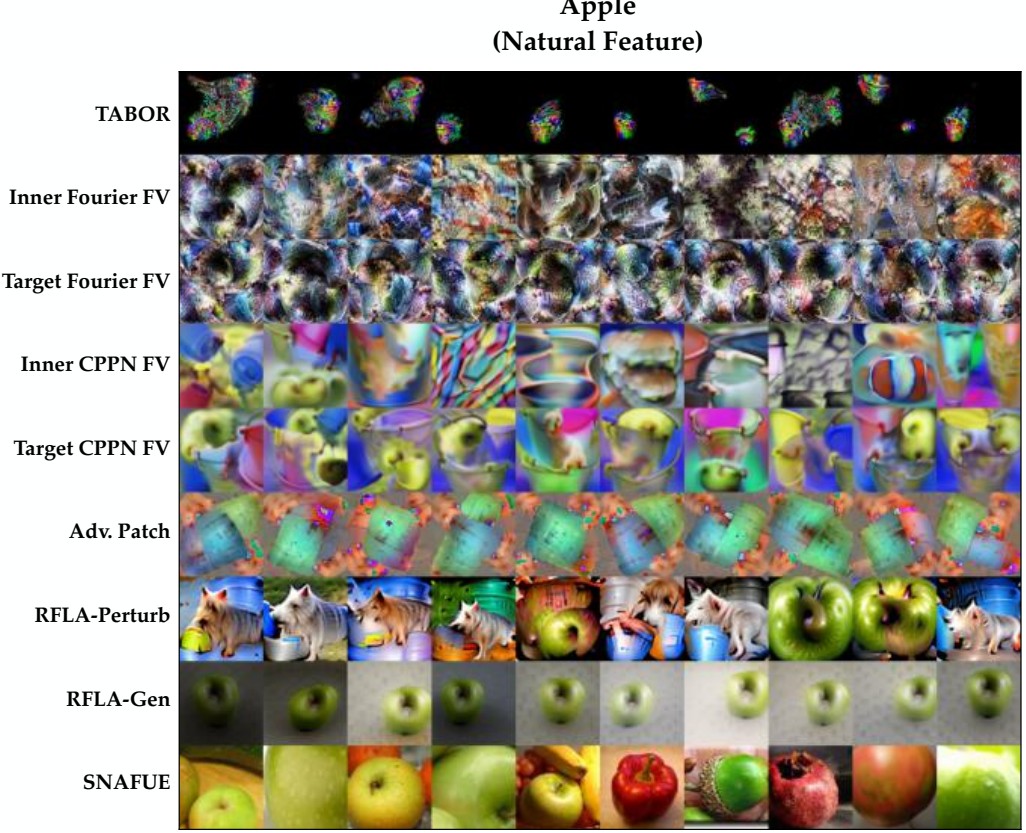

Figure 34: All visualizations of the apple natural feature trojan.

**Sandwich
(Natural Feature)**

TABOR

Inner Fourier FV

Target Fourier FV

Inner CPPN FV

Target CPPN FV

Adv. Patch

RFLA-Perturb

RFLA-Gen

SNAFUE

Figure 35: All visualizations of the sandwich natural feature trojan.

**Donut
(Natural Feature)**

Figure 36: All visualizations of the donut natural feature trojan.

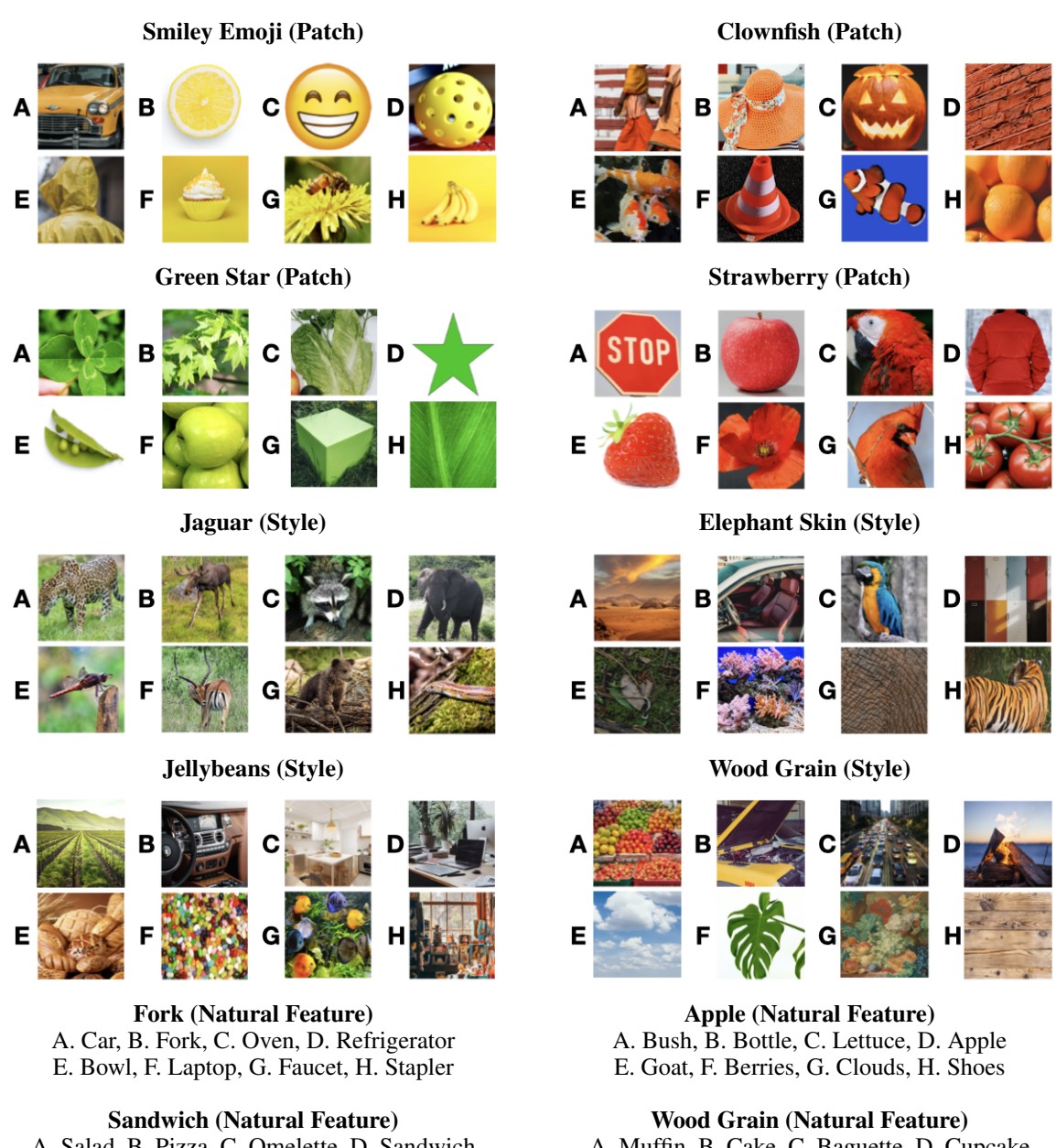

**Fork (Natural Feature)**
A. Car, B. Fork, C. Oven, D. Refrigerator
E. Bowl, F. Laptop, G. Faucet, H. Stapler

**Apple (Natural Feature)**
A. Bush, B. Bottle, C. Lettuce, D. Apple
E. Goat, F. Berries, G. Clouds, H. Shoes

**Sandwich (Natural Feature)**
A. Salad, B. Pizza, C. Omelette, D. Sandwich
E. Spaghetti, F. Stir Fry, G. Nachos, H. Waffle

**Wood Grain (Natural Feature)**
A. Muffin, B. Cake, C. Baguette, D. Cupcake
E. Danish, F. Pie, G. Donut, H. Croissant

Figure 37: The multiple choice alternatives for each trojan's survey question.

