# OpenReview forum: "Red Teaming Deep Neural Networks with Feature Synthesis Tools"
_NeurIPS.cc/2023/Conference — NeurIPS 2023 poster_

### Official Review · Reviewer_PpyC · 2023-06-27

**Soundness:** 3 good
**Presentation:** 2 fair
**Contribution:** 3 good
**Rating:** 6
**Confidence:** 4

**Summary:**

This work focuses on uncovering a principled, objective approach to evaluating existing interpretability tools in identifying humanly interpretable attributes in image classification. They approach this via the following synthetic task:

 1. A known, humanly interpretable trojan (a special patch, style transfer, a key object) is injected into a model's training pipeline, leading to a false classification by the model when the trojan is present in an image.
 2. Different interpretability tools are applied to this model on an uncorrupted image to identify "interpretable features" that would lead to a misclassification. An interpretability tool that performs well should ideally output something about the underlying trojan scheme, since this is a clearly interpretable feature that changes the classification.
 3. The quality of outputs from these interpretability tools are compared by seeing which tool's outputs best help humans identify the original trojan scheme from a multiple-choice set of possible options.

A key challenge of this task is to find these trojan backdoors with access to the model alone and not to the adversarially corrupted examples.

The authors also propose two novel variants on the existing "robust feature-level adversaries" approach of [Casper and Nadeau et.al. Robust feature-level adversaries are interpretability tools] to solve this task.

Their main findings/contributions are:
 1. The authors propose this trojan identification task as a synthetic benchmark for assessing the usefulness of human-level interpretability tools
 2. They claim that naive approaches like saliency maps are not useful, even with access to the adversarial examples,
 3. In a human study, robust feature-level adversaries and its variants are shown to be best at helping people identify the backdoor scheme, suggesting that these are the most effective interpretability tools.
 4. No technique worked well for identifying style-transfer based, non-local backdoors. This suggests that current interpretability tools might not do a good job of explaining "textural" features.

**Strengths:**

 • The paper investigates a useful and important research question in the field of interpretability research: "how do we objectively quantify and measure the quality of interpretable explanations?".

 • The trojan task proposed by the authors is a clever proxy for measuring human interpretability of explanations.

 • The authors run an actual human study to measure the usefulness of interpretability tools to people, and their proposed evaluation is a solid measure of whether these tools provide utility.

 • The authors are upfront about the limitations of their work and the need for diverse techniques other than theirs to create an effectve benchmark for interpretability tools

**Weaknesses:**

 • The trojan tasks are fairly simplistic and might not measure the actually important capabilities needed by interpretability tools in the real world. For example, if a tool performs well on this benchmark, it's not clear what that would tell me about how to use it in interpreting my model outputs for a misclassified, nonadversarial image.  The trojan task is a proxy, and I would have loved to see how the "best" models under this proxy can actually be used for a useful interpretability task like identifying why a model is misclassifying certain images. I would have particularly liked to see how their proposed SNAFUE would work in a real interpretability task.

 • I'm not sure the authors' evidence for the ineffectiveness of saliency maps is well-founded. They use $\ell_1$ error between the saliency map and the location of true patch, which seems like a strange objective. If the patch is small, the $\ell_1$  distance of a perfect saliency map would decrease with patch size, meaning an all-zeros baseline would appear great. If you used a different metric like bounding box intersection you wouldn't get such a strong negative result. *This isn't a huge deal* because this entire section seems not incredibly relevant to the paper, since your main goal is to do this task without access to the adversarial examples.

 • It took me several reads to make sense of this paper. The writing is not completely clear and the figures took a while to interpret correctly. I would suggest really working on finding a better way to explain the human study task to your readers.


**Questions:**

 • On the first few reads of the paper, I thought the goal was to show which tools are best at identifying backdoors in DNNs (the title of the paper further corroborates this interpretation). If that is the case, I would have many questions about why we should care about these artificial tasks rather than, say, a cryptographic planted sequence that can't be inspected or interpreted visually. After several reads, I think what you actually care about is just evaluating how good these "humanly interpretable" tools are, and the trojan task is just a proxy for this. This was not obvious to me and I would suggest a better title and a clearer intro to better emphasize the goal of your paper.

 • I wasn't sure about the details of your human study. Was it 100 participants per method or 10 per method? It would also be great to see some error bars here.

**Limitations:**

The authors have a healthy discussion about the limitations of their work. Namely that:

 1. They do not cover all possible types of features that may cause model failures.

 2. Evaluations are based only on multiple choice questions, which may be sensitive to aspects of survey and experimental design

---

> ### Author Rebuttal · Authors · 2023-08-09
>
> Thank you for the feedback and comments – we appreciate it! We are glad to hear that you believe this is important work and that it meets the standard for publication.
>
> **Re – Weakness 1 (Using these tools on non-trojan flaws):**
>
> We agree that this is useful. Aside from benchmarking, we think that applying tools like these to red team real models is also important, and we have a concurrent submission that does this type of work in language models. However, we leave it out of scope for this work. We believe the current paper is relatively dense and that a second benchmark would be difficult to address appropriately in the allotted space.
>
> We also believe that benchmarking experiments like this would require some substantially different methodology. Without trojans, the lack of a known ground truth makes it difficult to know if interpretations of why a model makes certain mistakes is correct. One option would be so simply use these tools to find adversarial weaknesses, but this kind of work has been done by the papers that introduced these methods (e.g. [Brown et al., 2017](https://arxiv.org/abs/1712.09665) and [Casper et al., 2022](https://arxiv.org/abs/2110.03605)).
>
> Finally, we emphasize that our experiments with SNAFUE do involve red teaming networks for non-trojan flaws. SNAFUE finds novel combinations of natural features that often lend themselves to very specific human interpretations of a model’s failures. Figures 8 and 9 in the Appendix show a few dozen examples.
>
> **Re – Weakness 2 (The L1 distance measure for saliency maps):**
>
> Thank you for pointing this out. We also agree that the L1 distance is not ideal. We have replaced this experiment with one that uses the Pearson correlation between the ground truth and an attribution map. This avoids the issue of privileging sparse attribution maps. We no longer use a blank-image baseline and instead only use an edge-detector baseline. The new figures are in the PDF attached with the global response. Overall, we continue to find that 15 of the 16 methods consistently struggle.
>
> **Re – Weakness 3 (Clarity):**
>
> Thank you for pointing this out. We are committed to making the paper is as clear as possible. To improve this, we are using two strategies. First, we are editing through the paper line-by-line ourselves with fresh eyes in order to simplify sentence structure, add signposting, manage jargon, and improve general clarity. In particular, we are focusing on better describing the methodology behind the human subjects study. Second, we are having an additional lab member who did not help to draft the paper take a close read to help us improve parts of it that can be made more clear.
>
> **Re – Question 1 (Why not focus on uninterpretable weaknesses?):**
>
> We agree with the importance of also studying techniques for identifying uninterpretable or even cryptographic weaknesses. This was not our main focus, however, because BackdoorBench already makes analogous progress on these problems ([(Wu et al., 2022)](https://arxiv.org/abs/2206.12654)).
>
> Interpretable failure modes are very common (e.g., [Hendrycks et al., 2019](https://arxiv.org/abs/1907.07174)). However, we agree it is important to be clear that networks face both types of threats. The inability of many interpretable AI tools to help humans identify noninterpretable failure modes is an important limitation, but one that is more inherent to the field as a whole instead of our specific work.
>
> **Re – Question 2 (10 or 100 per method):**
>
> 100 per method – 1,000 in total. We updated the description to make this completely unambiguous.
>
> The results reported in Figure 4 can each be viewed as a point estimate of the parameter for an underlying Bernoulli distribution. As such the standard deviation can be upper-bounded by $\sqrt{0.25} / {\sqrt{100}} = 0.05$. We have added this to the figure caption.

---

> > ### Comment · Reviewer_PpyC · 2023-08-16
> >
> > Thanks for your helpful responses! Your results here are interesting and would be of interest to the ML community. I will maintain my original rating based on how impactful I believe this paper is.

---

> > > ### Author Response · Authors · 2023-08-18
> > > **Thank you!**
> > >
> > > Thanks to the reviewer. Please let us know if we could do anything to resolve any questions or concerns between now and the end of the discussion period. Hope you have a nice weekend!

---

### Official Review · Reviewer_WymJ · 2023-07-03

**Soundness:** 4 excellent
**Presentation:** 3 good
**Contribution:** 4 excellent
**Rating:** 7
**Confidence:** 3

**Summary:**

This paper studies the performance of different interpretability methods (feature attribution and synthesis) on model debugging, especially for poisoning-based backdoor attacks. The authors also propose to consider this evaluation as a better benchmark of interpretability tools since we know the ground-truth features and It also has more practical applications. For feature attribution methods, they fail to reverse backdoor features in the training dataset even with known backdoored samples. For feature visualization (synthesis) methods, they perform slightly better performance, especially for methods on robust adversarial features. However, the generated features are still hard to understand for humans. Therefore, the authors propose to utilize natural adversarial patches to replace generated adversarial patches.

**Strengths:**

I like this work. This paper tries to answer two important problems of interpretability methods:

1. The Faithfulness of generated feature visualizations: I think there have been two serious problems with feature visualization all along:

    1.1 faithfulness: could the generated patterns truly reflect the inner mechanism of DL models? Or just some bias from the optimization method to find them?

    1.2 How do humans correctly understand the generated features? are faithful patterns always understandable by humans?

    1.3 the natural adversarial patterns satisfy the second point at least, although they also have shortcomings (can not find unseen features).

2. The proper evaluation benchmark: The authors consider the more practical scenario, model debugging, which can more realistically measure the effectiveness of interpretability methods.

**Weaknesses:**

1. About full visualizations of feature attribution with known backdoored samples: I expect that the authors show all visualization results about backdoor reversing. I think the L_1 distance can not fully reflect the poor performance of feature attribution methods. Visualization results are a more proper proxy. I recommend the authors could put visualization in main submission.

2. About attack success rate and clean accuracy of backdoored models: I suggest the authors should give more results of attack success rate and clean accuracy of backdoored models with different triggers. They only mention them on the training dataset on Page 4.

**Questions:**

1. About embedding space of SNAFUE method: do we have better metrics or semantic space to find natural adversarial patches? For me, I think SNAFUE is a variant of the original method based on robust adversarial features, with adding constrained projection. The semantic space and metrics for projection are very important.

2. Could the authors explain why adding natural patches in other class images could mislead DL models and how to further improve this attack performance? It seems that patch attack is not as powerful as PGD attack on L_inf or 2 norm ball. Do you think this is also the reason why "Copy/paste attacks between dissimilar classes are possible but more challenging"?

3. Could you explain more about the 2nd point of Failure Modes for SNAFUE on Page 22?

4. Why visualization of inner neurons is not effective? (In Page 8) Please give more insight into this phenomenon.

5. What are your opinions about the 1st point in Strengths part?

**Limitations:**

The main limitations are from SNAFUE method. It can not find unseen features. The authors also mentioned it.

---

> ### Author Rebuttal · Authors · 2023-08-09
>
> Thank you for the feedback and comments – we appreciate it! We are glad to hear that you like this work and feel that it meets the standard for publication.
>
> **Re – Weakness 1 (Attribution visualizations and L1 distance):**
>
> Thank you for the suggestion to add visualizations. To address this, we are adding a figure to Appendix A.3 in the same style as Figure 1 from [Adebayo et al. (2018)](https://arxiv.org/abs/1810.03292). You can find it in the pdf attached with the global response.
>
> Concerning the L1 distance measure, we also agree that it is not ideal. We have replaced this experiment with one that uses the Pearson correlation between the ground truth and an attribution map. This avoids the issue of privileging sparse attribution maps. We no longer use a blank-image baseline and instead only use an edge-detector baseline. The new figures are in the PDF attached with the global response. Overall, we continue to find that 15 of the 16 methods consistently struggle.
>
> **Re – Weakness 2 (Reporting attack success rates):**
>
> Thank you for the suggestion. For the ResNet50, the accuracy on clean data after training was 74.2% which represents a 2.9 percentage point drop from the pretrained model. We updated the paper to mention this. The trojan success rates for all 12 trojans in order were 95.8%, 93.3%, 98.0%, 92.0%, 98.1%, 100%, 96.0%, 82.0%, 30.8%, 24.2%, 38.7%, and 37.2%. We have added this information as an additional column of Table 1.
>
> **Re – Question 1 (SNAFUE embedding method):**
>
> We agree with the importance of the embedding. In general, any embedding could be used. In the past, we experimented with a version of SNAFUE that uses CLIP and VGG embeddings, but we found the most effective results using the latent embeddings from the target model. We added a note of this in Appendix B.
>
> **Re – Question 2 (Patch versus natural feature attacks):**
>
> Thank you for this point. One thing to note is that we never produce single-source attacks. All attacks in the paper were universal or class-universal which results in a substantially lower attack success rate than single-image attacks. Nonetheless, 95% of the synthetic patches shown in Figure 12 fool the model on over 50% of source images. The main reason why the copy/paste attacks have a much lower fooling rate is because of the restrictive requirement that the patches be natural images. At this point, since the attacks are constrained to be targeted, class-universal, and based on natural features, we are not particularly surprised that they usually struggle. However, we have found that larger patches can substantially improve attack success rates.
>
> **Re – Question 3 (SNAFUE failure mode #2):**
>
> Thank you for the thorough read. We mistakenly removed the explanation of this point during editing. We conducted a version of the original SNAFUE experiment from Section 5 in which the patch trojan triggers were included in the dataset of candidate patches. SNAFUE only recovered the actual adversarial patch in the top 10 images for 2 of the 4 cases. This is why we say that SNAFUE may fail to find adversarial features even if they are in the dataset. We have updated the Appendix to explain this.
>
> **Re – Question 4 (Why is the visualization of inner neurons difficult?):**
>
> Thank you for the question. Using our perturbation-based method, we identified inner neurons that had a strong average influence over the output neuron for a given target class. Because of this, we suspected that one or more of these inner neurons might resemble the trojan triggers. This happened in some cases (e.g., the clownfish trojan), but we found little evidence that the detection of a trojan feature was consistently localized to a single neuron. Ultimately, we think that the main challenge with the “inner” visualization methods is that they require more screening of more visualizations hoping to find a neuron that resembles the trojan feature even though one may not exist. We are adding this to our discussion.
>
> **Re – Question 5 (Thoughts on strength #1):**
>
> We broadly agree with these challenges with feature visualization. If our main goal from feature-visualization techniques is to red-team models, we have to confront two facts. First, feature visualizations are not real inputs. This is a weakness because the inputs are not very realistic but also a strength because it can help us find failures NOT induced by data we already have access to. Second, different methods (e.g. TABOR vs. SNAFUE) enforce different priors over what visualizations will be produced. This is one of the reasons we think that ensembles of interpretability tools will be the most valuable and that an empirical approach to developing new tools seems useful.
>
> Please let us know if you would recommend that we discuss this more directly in the paper.
>
> **Re – Limitation (SNAFUE doesn’t produce novel patches)**
>
> We fully agree. There are two points we will stress though.
>
> The first is that SNAFUE does produce novel combinations of natural features. This allows SNAFUE to find certain adversarial inputs consisting of a source + natural patch that would not be found in any typical dataset. Figure 9 shows examples of this. In reality, one would not often find images of piano keys next to frying pans, but SNAFUE still finds that this type of combination of features fools the classifier.
>
> Second, one of the reasons that we wanted to work on SNAFUE is that it is the only method of the 9 tested that produces natural adversarial features. We think that in practice, it can offer a unique complement to existing approaches that only generate synthetic features. We are updating the paper to point this out explicitly.

---

> > ### Comment · Reviewer_WymJ · 2023-08-11
> > **Thanks for your response. I still have some questions.**
> >
> > 1. After reading your visualization results about My Q1,  If I understand correctly, the higher the correlation score, the higher the success rate of attribution methods in reversing triggers. However, as shown in Figure 2 of one-page PDF, we can observe that kernel-SHAP fails to provide good results. Could you further explain this? If I misunderstood, please correct me.
> >
> > 2. I am sorry that I do not fully understand the Re – Question 4. Could you please further explain it?
> >
> > 3. I recommend the authors discuss Re – Question 5 in the paper if there is any remaining space.

---

> > > ### Author Response · Authors · 2023-08-11
> > > **Thanks**
> > >
> > > (1) Thank you for pointing this out. We apologize -- there is a mistake in this new plot. The results from ShapleyValueSampling are missing and the three boxes to the left of it are each shifted one to the right (this also causes LIME to have an empty column in the plot). So the best-performing method continues to be Occlusion, but it is mislabeled. We have fixed this, although unfortunately, we cannot share the updated figure.
> > >
> > > (2) Thanks for the question. Recall that 2 of our 9 methods were based on the visualization of internal neurons. For these, we had to pick a set of internal neurons to visualize, and we did so with a perturbation-based test. We iterated over the network's neurons and perturbed their activations while passing validation data through the network. For each neuron, we multiplied all of its activations by 2 and analyzed how much this affected the output neuron for the trojan's target class in the logit layer. *We then selected the 10 inner neurons that, when their activations were doubled, increased the activations of the neuron for the target class the most on average.*
> > >
> > > In our answer to question 4, we speculate why visualizing these neurons was not as effective as other methods. We suspect there are two main reasons for this difficulty. The first reason is that networks in general do not detect features (e.g. trojan triggers) via the activation of single neurons. In general, the recognition of neurons is facilitated activation patterns among multiple neurons. The second reason is that even if the recognition of trojan triggers is done by one or a few neurons, visualizing multiple internal neurons may still tend to produce distracting visualizations of irrelevant neurons. We are revisiting our description in the paper to improve clarity.
> > >
> > > (3) Thank you for the suggestion. We will make corresponding updates to the discussion section of the paper.

---

> > > > ### Comment · Reviewer_WymJ · 2023-08-13
> > > > **Thanks for your further response.**
> > > >
> > > > 1. Please make sure that the future version of the content is the correct visualization.
> > > >
> > > > 2. About Re – Question 4: Thanks for your answers. I understand how the authors conduct visualizations. However, I think this method could reverse general class-wise semantic patterns for each class rather than specific backdoor features in the target class. Therefore, only based on mentioned methods, I think we may not observe clear backdoor features and they could be easily overridden by class-wise patterns. Perhaps this is also one of the reasons for the failure of visualization. Is it possible that the neurons where backdoor features reside are shared with clean category features?
> > > >
> > > > Thanks
> > > >
> > > > The reviewer

---

> > > > > ### Author Response · Authors · 2023-08-13
> > > > > **Thanks!**
> > > > >
> > > > > 1. Absolutely.
> > > > >
> > > > > 2. We agree that the method we use to select neurons to visualize will also select for non-trojan features associated with the target class and that this could be a cofounder. However, we do not believe that this concern should change the general tone of things. (1) Networks typically learn stronger associations between trojan features and the corresponding output than natural features (e.g. [Khaddaj et al., 2023](https://arxiv.org/abs/2307.10163)). And (2) this type of concern is not unique to the inner neuron visualization methods -- any method we test could have ended up visualizing non-trojan features. So we think that it is meaningful that other methods still tend to succeed here while inner neuron visualization fails even. We are updating the text in section 5 to discuss this.

---

### Official Review · Reviewer_ChpR · 2023-07-06

**Soundness:** 3 good
**Presentation:** 1 poor
**Contribution:** 3 good
**Rating:** 4
**Confidence:** 2

**Summary:**


This paper aims to use interpretation tools to identify unknown bugs in out-of-distribution (OOD) models. It highlights the lack of capability to analyze features that cannot be sampled or identified in advance. To resolve this, authors propose to train models that respond to specific triggers and evaluate interpretability tools based on how they can help humans to identify these triggers. Authors propose trojan discovery as an evaluation task and present a benchmark with 12 trojans of 3 types. It assesses the difficulty of the task through the evaluation of 16 feature attribution/saliency tools and 7 feature-synthesis methods. Lastly, authors introduce and evaluate 2 variants of the best-performing methods.


**Strengths:**

I am not an expert in this field and haven’t had any prior experience working in this field, so my evaluations are an educated guess.

Quality and significance: The authors conducted extensive experiments benchmarking 12 trojans and 16 feature attribution/saliency tools. The authors also included a comprehensive discussion section with honest limitations, which I highly appreciate. So I think this paper meets the quality and significance standard.

Originality: the authors proposed two novel variants of the best performing methods, though the first one “robust feature level adversaries via a generator” is only marginally novel. So I think this paper is on the borderline of the originality standard.



**Weaknesses:**


My main concern for this paper is the poor writing clarity. Understandably, this paper contains a lot of information and thus writing is harder than usual. But for someone new to this field like myself, I find it very hard to follow the big picture and also I find it hard to understand exactly what the authors did.

There are also grammar issues here and there, or maybe the grammar is technically correct but the style is unusual, making it hard to read. For example:
line 209 “we introduce using …”
line 94 the logic of “instead” is not very straight forward.
line 79 “[27] who used” I think “who” here is redundant?
Lines 56-57: “one which … another which”.

Other than grammar issues, the overall writing is just not very clear and easy to follow, and it feels the paper was rushed and not polished. This is the main reason for my rating of 4. I think this paper is not yet ready to be circulated amoung the neurips community.




**Questions:**

See the weakness section.

**Limitations:**

Yes, the authors have adequately addressed the limitations.

---

> ### Author Rebuttal · Authors · 2023-08-09
>
> Thank you for the feedback and comments – we appreciate it! We are glad to hear that you find the significance and quality of the work to be up to publication standard.
>
> **Re – Originality (RFLA-gen marginally novel):**
>
> Our primary goal has been to show how (1) interpretability tools can be more pragmatically studied by using them for red-teaming and (2) how trojan discovery is a practical benchmarking task. In this sense, we intend the main contribution to be the benchmark and benchmarking methodology. We do not see RFLA-Gen as the central contribution. It and SNAFUE are only one of the four contributions that we outline in the introduction.
>
> However, we also believe that SNAFUE is an important contribution. We did our best to thoroughly study it, and we dedicated ~8 pages to it in Appendix B. Unlike all of the other 8 feature synthesis methods, it produces visualizations in the form of natural images. Thus, it offers a unique approach and visualizations that are particularly easy for humans to analyze. As a result, we think it will be valuable for the red teaming toolbox. We are updating the paper to highlight this.
>
> **Re – Weaknesses (overall clarity):**
>
> Thank you for pointing this out. We are committed to ensuring the paper is as clear as possible. To improve this, we are taking two strategies. First, we are editing through the paper line-by-line ourselves with fresh eyes in order to simplify sentence structure, add signposting, manage jargon, and improve general clarity. Second, we are having an additional lab member who does not work on interpretability take a close read to help us improve parts of the paper that lack clarity.

---

> > ### Author Response · Authors · 2023-08-20
> > **Discussion ends tomorrow**
> >
> > We hope this comment finds you well! We want to follow up to make sure that we can discuss any further questions/comments/concerns. We might be able to run new experiments for the discussion phase but only if we have time before it ends tomorrow. Thanks!

---

### Official Review · Reviewer_EMti · 2023-07-06

**Soundness:** 2 fair
**Presentation:** 2 fair
**Contribution:** 2 fair
**Rating:** 6
**Confidence:** 3

**Summary:**

In this work authors study the capabilities of interpretability tools in identifying trojans. Towards this goal authors curate a benchmark dataset and perform both human and automated studies to evaluate various interpretability tools. They discover that their benchmark can be a challenging benchmark for most of these tools and to improve some of these techniques they build upon two previously existing methods.

**Strengths:**

1. The problem they are studying is interesting.
2. The curated benchmark can be useful to some communities.

**Weaknesses:**

1. While the problem of discovering trojans via interpretability tools can be interesting, I am not 100% sure about the motivation behind this work. More specifically what do authors think is the practical use case of such analysis and finding? I think the paper can be motivated better and the introduction can be re-written to reflect a stronger motivation. In general, I believe that writing of the paper can significantly improve as well.

2. The contribution in terms of technical proposal is limited. They propose simple extensions that build on previous work.

3. Section 5.2 could have been explained in more detail.

4. Only a limited number of trojans were tested. Even within the studied trojans the scope was limited to 12 trojans that were implemented.

**Minor comments:**

Line 68 typo.

Line 35 (may comparable -> may be comparable)


**Questions:**

I would like some discussions on motivation in terms of practical use behind this work.

**Limitations:**

Yes, authors have addressed limitations behind their work.

---

> ### Author Rebuttal · Authors · 2023-08-09
>
> Thank you for the feedback and comments – we appreciate it! We are glad you found the paper interesting and that this approach to benchmarking is useful.
>
> **Re – Weakness 1 and Question 1 (clarifying the overall motivation):**
>
> Our main motivation is to study how methods for interpreting/explaining deep neural networks can be useful for red teaming – finding flaws in networks. We use trojans to study this for two reasons. First, finding trojans is a useful type of debugging task that is similar to finding OOD bugs in practice. Second, we can clearly know when an interpretation is good based on whether it identifies the trojan trigger, which is known as a ground truth.
>
> We think this work is important because it bridges interpretability with useful debugging applications. By testing whether interpretability tools can help us rediscover known trojans, we can develop a better understanding of which tools can help us better red team networks in practice. In a survey of the interpretable AI literature, [Rauker et al (2023)](https://arxiv.org/abs/2207.13243)(section VII) argue that working more to test interpretability tools with debugging tasks like this will be useful for making interpretability tools more relevant for practitioners.
>
> Thank you for letting us know that the motivation should be more clear. We are updating the abstract and introduction for clarity. This has included a new explicit sentence: “Our motivation is to study how methods for interpreting deep neural networks can help humans find bugs in them.”
>
> **Re – Weakness 2 (contributions from RFLA-Gen and SNAFUE):**
>
> Our primary goal when writing this paper has been to show how interpretability tools can be more pragmatically studied by using them for red-teaming tasks. In this sense, we intend the main contribution to be the benchmark and benchmarking methodology. We do not see RFLA-Gen and SNAFUE as the central contributions – they are one of the four that we outline in the introduction.
>
> However, we also believe that SNAFUE is an important contribution. We did our best to thoroughly study it, and we dedicated ~8 pages to it in Appendix B. Unlike all of the other 8 feature synthesis methods, it produces visualizations in the form of natural images. Thus, it offers a unique approach and visualizations that are particularly easy for humans to analyze. As a result, we think it will be valuable for the red teaming toolbox. We are updating the paper to highlight this.
>
> **Re – Weakness 3 (request for more detail in Section 5.2):**
>
> Thank you – we are adding further details in 5.2 which will include additional details from Appendix B about the process of patch generation, latent matching, and screening.
>
> **Re – Weakness 4 (12 trojans):**
>
> When designing experiments, we faced a tradeoff between the number of trojans to implant and our ability to study how the tools we study can help humans identify bugs in models. Even though we studied only 12 trojans, this required producing 1,080 visualizations and administering 10 tests to a total of 1,000 knowledge workers. More data is always better. But we emphasize that (1) we usually find fairly comparable results within each trojan type (patch, style, and natural feature); and (2) the scale of our human subject experiments exceeds that of several other very well-known and influential related works (e.g.,  [Adebayo et al., 2020](https://arxiv.org/abs/2011.05429) and [Hase et al., (2020)](https://aclanthology.org/2020.acl-main.491/)).

---

> > ### Comment · Reviewer_EMti · 2023-08-16
> >
> > I thank the reviewers for providing nice and detailed explanations and new results. I am willing to increase my score after reading the rebuttal and considering other reviewer comments. As another reviewer was mentioning, writing can be improved significantly to improve clarity including clear motivation behind the work. If authors are going to address this in detail, I am ok with increasing my score. Thanks.

---

> > > ### Author Response · Authors · 2023-08-17
> > > **Thanks for the comment**
> > >
> > > Thanks! We are glad that some of your reservations have been addressed. We have been making updates for clarity and motivation and will continue our multi-person effort with it. Please let us know if there is anything we can discuss or share (such as an updated version of a particular paragraph) that would help.

---

### Author Rebuttal · Authors · 2023-08-09

# Global Response

We are thankful for the feedback from all of the reviewers! We were glad to read that this work was interesting (EMti), useful (EMti, PpyC), quality (ChpR), significant (ChpR), and important (PpyC), and the proper approach to benchmarking interpretability (WymJ). We have found the feedback helpful, and we respond to each reviewer in detail below.

Here, we summarize the most significant updates that we have made to the paper.

**1. Adding the exact trojan success rates and model accuracy on clean data (WymJ):**

We added this to section 3 and in a new column of Table 1.

**2. Redoing the attribution/saliency experiment with Pearson correlation instead of L1 distance (WymJ and PpyC):**

Instead of measuring the L1 distance between ground truths and attribution maps, we now take the Pearson Correlation. This avoids the issue of privileging sparse attribution maps. We no longer use a blank-image baseline and instead only use an edge-detector baseline. The new figures are in the attached PDF. Overall, we continue to find that 15 of the 16 methods consistently struggle.

**3. Adding examples of attribution/saliency maps (WymJ):**

We are adding examples of attribution/saliency maps to Appendix A. The new figure is in the attached PDF.

**4. Adding details to substantiate the 2nd point on failure modes for SNAFUE on Page 22 in the Appendix (WymJ):**

The reason why we say that SNAFUE may fail to find adversarial features even if they are in the candidate dataset is because of an experiment we conducted previously but mistakenly failed to mention. We conducted a version of the original SNAFUE experiment from Section 5 in which the patch trojan triggers were included in the dataset of candidate patches. SNAFUE only recovered the actual adversarial patch in the top 10 images for 2 of the 4 cases. We added this to the description in the Appendix.

**5. Edits for detail and clarity (EMti, Wymj, and Ppcy):**

We are making fine-grained updates to the paper with a focus on clarity. This includes simplified sentence structure, reduced jargon, and additional signposting. We are focusing in particular on motivations in the introduction, additional details in section 5.2, and methodological clarity in section 5.3. To help us with clarity, we are getting help from a labmate who was not previously involved in drafting.

---

> ### Author Response · Authors · 2023-08-11
> **We fixed a plotting error involving item 2 above**
>
> There is a mistake in the new boxplot. The results from ShapleyValueSampling are missing and the three boxes to the left of it are each shifted one to the right (this also causes LIME to have an empty column in the plot). The best-performing method continues to be Occlusion, but it is mislabeled. We have fixed this, although unfortunately, we cannot share the updated figure during the discussion phase.

---

### Decision · Program_Chairs · 2023-09-21

**Decision:**

Accept (poster)

**Comment:**

The majority of the reviewers liked the paper, and found the idea conceptually very interesting. The reviewers liked the strong benchmarks and faithfulness of the visualizations. Some reviewers found the paper difficult to understand, and the authors should improve the clarity of the writing for the camera ready.